# Orexin-A and endocannabinoids are involved in obesity-associated alteration of hippocampal neurogenesis, plasticity, and episodic memory in mice

Nicola Forte [1,12], Serena Boccella[2,12], Lea Tunisi[1,12], Alba Clara Fernández-Rilo[1], Roberta Imperatore[3], Fabio Arturo Iannotti[1], Maria De Risi[4,5], Monica Iannotta[2], Fabiana Piscitelli[1], Raffaele Capasso [6], Paolo De Girolamo[7], Elvira De Leonibus [4,5], Sabatino Maione[2,8], Vincenzo Di Marzo[1,9,10,11✉] & Luigia Cristino [1✉]

The mammalian brain stores and distinguishes among episodic memories, i.e. memories formed during the personal experience, through a mechanism of pattern separation computed in the hippocampal dentate gyrus. Decision-making for food-related behaviors, such as the choice and intake of food, might be affected in obese subjects by alterations in the retrieval of episodic memories. Adult neurogenesis in the dentate gyrus regulates the pattern separation. Several molecular factors affect adult neurogenesis and exert a critical role in the development and plasticity of newborn neurons. Orexin-A/hypocretin-1 and downstream endocannabinoid 2-arachidonoylglycerol signaling are altered in obese mice. Here, we show that excessive orexin-A/2-arachidonoylglycerol/cannabinoid receptor type-1 signaling leads to the dysfunction of adult hippocampal neurogenesis and the subsequent inhibition of plasticity and impairment of pattern separation. By inhibiting orexin-A action at orexin-1 receptors we rescued both plasticity and pattern separation impairment in obese mice, thus providing a molecular and functional mechanism to explain alterations in episodic memory in obesity.

[1] Institute of Biomolecular Chemistry, Consiglio Nazionale delle Ricerche (CNR), Pozzuoli, NA, Italy. [2] Department of Experimental Medicine, Division of Pharmacology, University of Campania Luigi Vanvitelli, Napoli, Italy. [3] Department of Science and Technology, University of Sannio, Benevento, Italy. [4] Telethon Institute of Genetics and Medicine, Pozzuoli, Naples, Italy. [5] Institute of Biochemistry and Cell Biology, Consiglio Nazionale delle Ricerche (CNR), Monterotondo Scalo, Rome, Italy. [6] Department of Agricultural Sciences, University of Naples Federico II, Portici, NA, Italy. [7] Department of Veterinary Medicine and Animal Productions, University Federico II, Napoli, Italy. [8] I.R.C.S.S., Neuromed, 86077 Pozzilli, Italy. [9] Heart and Lung Research Institute of Université Laval, Québec City, QC, Canada. [10] Institute for Nutrition and Functional Foods, Centre NUTRISS, Université Laval, Québec City, QC, Canada. [11] Canada Excellence Research Chair on the Microbiome-Endocannabinoidome Axis in Metabolic Health, Université Laval, Québec City, QC 61V0AG, Canada. [12] These authors contributed equally: Nicola Forte, Serena Boccella, Lea Tunisi. ✉email: vincenzo.di-marzo.1@ulaval.ca; luigia.cristino@icb.cnr.it

Episodic memories are associated with the personal experience of everyday life, and the ability to distinguish between two similar past events is known as pattern separation (PS). This ability is regulated by the hippocampus, where inputs from the entorhinal cortex are transformed into distinct and relatively non-overlapping representations in the CA3 area. This area works as an associative field of the inputs coming from the dentate gyrus (DG)[1,2]. PS is altered in different chronic pathologies, such as Alzheimer's disease[3] and multiple sclerosis[4]. Although obesity is known to be associated with a dysfunction in episodic memory formation and retrieval[5–7], which underlies maladaptive decision making for food intake and choice in humans[8–10], this phenomenon has not yet been thoroughly studied from the mechanistic point of view.

Adult neurogenesis, a structural form of neural plasticity that contributes to the maintenance of brain functions with aging[11], takes place in the DG, subventricular zone, and olfactory bulb, and it is strictly implicated in the regulation of PS, which is impaired when newborn neurons are ablated by a low dose of X-irradiation[12]. On the contrary, potentiation of adult hippocampal neurogenesis (AHN by genetic manipulation improves cognitive performance during distinction tasks between two similar contexts[13,14].

Within the DG, neurogenesis mediates a specific form of LTP[15–17] known to play a substantial role in PS. Deletion of NR2B-containing NMDA receptors in newborn neurons contributes, per se, to LTP and PS inhibition[18,19]. On the other hand, LTP in the DG increases AHN[20,21]. Positive lifestyle changes, such as physical activity and caloric restriction, are beneficial by promoting hippocampus-related cognition through enhancing AHN in rodents[22,23]. The anorexigenic hormone leptin enhances AHN and synaptic function, which facilitates spatial learning and memory function[23], suggesting a neurotrophic role of leptin within the hippocampus. Accordingly, AHN is altered following prolonged consumption of a diet with a high-fat content (HFD) and in obese leptin receptor knockout (db/db) mice[24–27], consistent with impairment in hippocampal-dependent behavioral tasks and paradigms in these mice[28]. On the other hand, very little is known about adult neurogenesis in the hippocampus of ob/ob obese mice concerning the regulation of PS.

Endocannabinoids (eCBs) promote and regulate AHN[29,30] and neuronal progenitors produce eCBs[31] and possess a functional eCB system, including eCB major targets, the cannabinoid CB1 and CB2 receptors[32,33]. A direct role of eCBs has been proposed in the formation of episodic memory[5,34] whereas *Cannabis* consumption may interfere with the formation of new memories[35]. The biosynthesis of the eCB, 2-arachidonoylglycerol (2-AG) via diacylglycerol lipase α (DAGLα), and subsequently the signaling of CB1 receptors, are stimulated by OxA[36], a neuropeptide produced by neurons in the lateral perifornical area of the hypothalamus (LH) and involved in the sleep/wake cycle, arousal, appetite, and energy homeostasis and acting via the $G_{q/11}$ protein-coupled orexin receptor-1 (Ox1-R) and orexin receptor-2[37,38]. OxA is emerging as a regulator of neurogenesis and hippocampal plasticity[38–41]. We previously reported that OxA neurons in the LH of obese mice undergo a rewiring from glutamatergic to GABAergic afferents and subsequent 2-AG/CB1 receptor-mediated disinhibition, which triggers, in turn, excessive OxA signaling to LH output areas[42,43].

Here we investigated the role of the OxA/Ox1-R/DAGLα/2-AG/CB1 signaling cascade in the regulation of AHN-dependent LTP and PS in adult HFD and ob/ob mice. In particular, we analyzed the number of immature dentate granule cells (GCs), identified by the stereotypic pattern of doublecortin-immunoreactive (DCX-ir) expression (for review, see Lazarov et al.[44], Kempermann et al.[45]) in the bottom third of the DG of

the hippocampus, including the subgranular layer (SGL) and inner and medial molecular layers (IML-MML) of adult obese mice (i.e. leptin knockout ob/ob mice and wild-type ob gene expressing homozygous siblings fed with HFD), in comparison to age-matched lean mice as control (i.e. wild-type ob gene mice fed with regular chow with standard fat content). In this work, we report data that collectively support the existence of a causal link between memory dysfunction and orexin and endocannabinoid signaling crosstalk in obesity.

## Results

**PS is altered in obese mice.** Impairment of cognitive and memory tasks has been demonstrated in obese animals and humans[46]. To test if HFD and ob/ob mice present impairment in discriminating stored memories as effects of a deficit in PS, the novel object recognition (NOR) task was performed[14] after having characterized the lack of impact of obesity on baseline motor abilities. Based on the evidence that obese mice are more sensitive to the behavioral responses during spontaneous behavior or ethologically based tests[47], the light−dark box test and open-field test were performed to exclude the influence of the obese phenotype on general motor functioning (Supplementary Fig. 1). In the light−dark box test, ob/ob mice showed a reduction in the number of transitions between light and dark chambers and spent less time in the light box and showed a longer latency of the first entry in the dark chamber, while HFD showed a reduction in the time spent in the light box and a reduction in latency ($n = 10$ mice/group; Transition: lean $= 45.9 \pm 3.5$, HFD $= 52.10 \pm 2.1$, ob/ob $= 11.8 \pm 1.9$, ANOVA test and Bonferroni post hoc test, ***$p < 0.001$, F $= 67.72$; time in light: lean $= 387.9 \pm 16.8$ s, HFD $= 258.4 \pm 25.6$ s, ob/ob $= 115.1 \pm 18.6$ s, ANOVA test and Bonferroni post hoc test, ***$p < 0.001$, F $= 43.3$; Latency: lean $= 24.90 \pm 5$ s, HFD $= 4.8 \pm 0.5$ s, ob/ob $= 71.90 \pm 3.9$ s, ANOVA test and Bonferroni post hoc test, ***$p < 0.001$ and *$p < 0.05$, F $= 87.02$) (Supplementary Fig. 1a−c).

During the open-field test (Supplementary Fig. 1d), compared with lean and HFD mice, ob/ob mice traveled less distance ($n = 10$ mice per group. Traveled distance: lean $= 43 \pm 5.5$ m, HFD $= 46.46 \pm 3.7$ m, ob/ob $= 10.04 \pm 1.4$ m, ANOVA test and Bonferroni post hoc test, **$p < 0.01$, F $= 23.57$) and spent less time in the central zone ($n = 10$ mice per group. Time in center: lean $= 29.6 \pm 4.1$ s, HFD $= 26.51 \pm 2.6$ s, ob/ob $= 0.4 \pm 0.1$ s, ANOVA test and Bonferroni post hoc test, ****$p < 0.0001$, F $= 32.34$) (Supplementary Fig. 1e, f). The ob/ob mice also significantly decreased their entries into the central zone as compared with lean mice (Supplementary Fig. 1g) (visit to central area: lean $= 26.8 \pm 3.7$, HFD $= 30.80 \pm 3.7$, ob/ob $= 0.4 \pm 0.1$, Kruskal−Wallis test and post hoc Dunn's test, ***$p < 0.001$, Kruskal−Wallis statistic $= 19.93$). Altogether, these findings highlight the presence of an anxiety-like behavior in ob/ob mice, which was prevented by extending the habituation period to 2 weeks (traveled distance $= 24.71 \pm 1.3$ m; time in center $= 10.4 + 0.9$ s; visit to central area $= 12.70 \pm 1.4$ s, Wilcoxon test, **$p < 0.01$) (Supplementary Fig. 1h−k).

To test PS in obese mice we selected two behavioral tasks that do not require to food deprive the animals and that use appetitive reward to motivate them. Both tasks however have been previously shown to be sensitive to alteration in DG neurogenesis[14,48,49]. As the first task, we used a modified version of the novel object recognition (NOR) task, in which animals are tested for their capability of distinguishing a novel vs a familiar object, with high (same color, similar shape) and low (different color, different shape) degree of similarity. The test was performed at 1.5- and 24-h retention intervals (Fig. 1a). During the sample phase, mice were always exposed to two identical objects (Fig. 1b). In this condition,

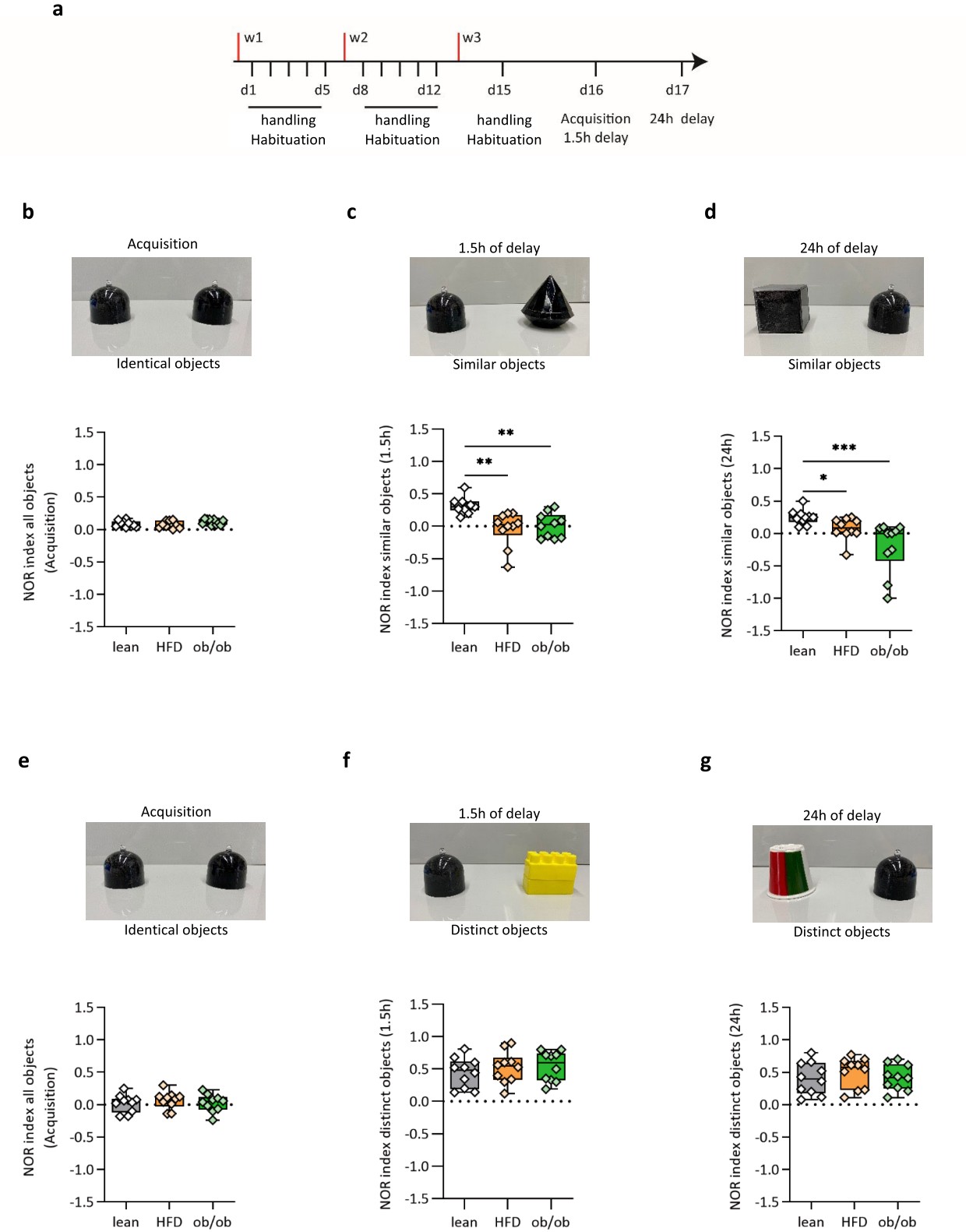

we found a very similar time of exploration for lean, HFD, and ob/ob mice during the study phase (NOR-discrimination index: lean = 0.08 ± 0.04, HFD = 0.07 ± 0.05, ob/ob = 0.1 ± 0.04, $n = 10$ mice per group, Fig. 1b). These data suggest that the animals do not show a bias towards the right or left side. Novel object recognition was tested by substituting the new object with a similar object with the same color, but different shape. Obese mice showed a significant reduction in the discrimination index as compared to lean mice (NOR-discrimination index: lean = 0.31 ± 0.03, HFD = −0.04 ± 0.08, ob/ob = 0.01 ± 0.05, $n = 10$ mice per group; Kruskal−Wallis test and post hoc Dunn's test, **$p < 0.01$, Kruskal−Wallis statistic = 15.07) (Fig. 1c), indicating that both HFD and ob/ob mice did not discriminate between the two similar objects at 1.5 h after the study phase. The same mice were tested 24 h after the

**Fig. 1 Obesity impairs pattern separation in the NOR test. a** Representative scheme and timeline of the novel object recognition test in lean, HFD, and ob/ob mice; $n = 10$ per group. **b** NOR in lean, HFD, and ob/ob mice calculated during the habituation phase; $n = 10$ per group. **c** Short-term retention phase using similar objects; NOR in lean, HFD, and ob/ob mice calculated after 1.5 h, NOR $= -0.04 \pm 0.08$ and $0.01 \pm 0.05$ in HFD and ob/ob respectively compared to lean mice with a NOR $= 0.31$; $n = 10$ per group, Kruskal−Wallis test and post hoc Dunn's test, $**p < 0.01$, Kruskal−Wallis statistic $= 15.07$. **d** Long-term retention phase. NOR in lean (gray), HFD (orange), and ob/ob (green) mice calculated after 24 h, NOR $0.25 \pm 0.03$ in lean NOR $= 0.07 \pm 0.05$ in HFD and NOR $= -0.20 \pm 0.08$, 0.1 in ob/ob; $n = 10$ mice per group. Kruskal−Wallis test and post hoc Dunn's test, $*p < 0.05$ and $***p < 0.001$, Kruskal−Wallis statistic $= 16.54$. **e−g** Short-term retention phase using distinct objects; NOR in lean, HFD, and ob/ob mice calculated in habituation phase after 1.5 and 24 h for the long-term retention phase, respectively; $n = 10$ per group. The box plots elements are: center line, median (Q2); square symbol, mean; box limits, 25th (Q1)−75th (Q3) percentiles; whisker length is determined by the outermost data points. Source data are provided as a Source Data file.

sample phase by presenting to them again the familiar object as well as a further novel similar object; the relative position of the novel and familiar objects was switched in comparison to the 1.5 h test (Fig. 1d). During this long-term retention test, there was a significant difference between obese and lean mice. Under presentation of similar novel objects, lean mice remembered the familiar object significantly better than obese (NOR-discrimination index: lean $= 0.25 \pm 0.03$, HFD $= 0.07 \pm 0.05$, ob/ob $= -0.20 \pm 0.12$, $n = 10$ mice per group; Kruskal−Wallis test and post hoc Dunn's test, $*p < 0.05$ and $**p < 0.01$, Kruskal−Wallis statistic $= 16.54$) (Fig. 1d). As a control experiment, a separate set of animals was tested in the classical version of the NOR, by using two completely different (in color and shape) objects during the test phases. Under these conditions, no defect was observed in obese mice, at both 1.5- and 24-h retention intervals (Fig. 1e−g).

To further confirm that HFD mice manifested impaired PS ability, we used a modified version of the water maze task that has been previously reported to detect specific navigational ability improvement or impairment as a consequence of exercise training-induced adult neurogenesis or genetic constitutive ablation (Cyclin D2 knockout-Ccnd2−/− mice) of adult neurogenesis in mice, respectively[48] (Fig. 2a). We found that HFD did not affect the use of beacon navigation to reach a visible platform in the water maze (Fig. 2b, c). Similarly, HFD did not impair the ability of mice to reach the platform when they were submerged in the North quadrant (days 2–4), as evidenced by a day-dependent decrease in the path length and latency to reach it in both groups ($n = 6−9$ mice per group; ANOVA test and Bonferroni post hoc test; latency $*p < 0.0001$; F $= 24.37$; path length $*p < 0.0001$; F $= 22.89$, Fig. 2b, c). Consistently, in the probe trial, when the platform was removed and animals were allowed to search for it for 60 s, all groups showed a clear preference for the correct goal quadrant (lean: target quadrant $= 44.29\% \pm 4.1\%$, other quadrants $= 18.57 \pm 1.3$; HFD: target quadrant $= 51.72\% \pm 4.3\%$ and other quadrants $= 16.07\% \pm 1.4\%$; $n = 6−9$ mice per group; ANOVA test and Bonferroni post hoc test, $*p < 0.0001$; F $= 56.9$, Fig. 2d, e). However, a deeper analysis showed that HFD, compared to lean mice had a lower number of goal crossings ($n = 6−9$ mice per group; $\#p < 0.05$; F $= 4.82$; lean $= 10 \pm 0.8$, HFD $= 7.5 \pm 0.7$) (the specific annulus where the platform was located) and a reduced path efficiency (lean $= 0.5 \pm 0.08$, HFD $= 0.3 \pm 0.05$; $\#p < 0.05$; F $= 5.55$), associated to increased latency to the first entry in the goal (lean $= 7.9 \pm 2.1$, HFD $= 18.21 \pm 3.7$; $n = 6−9$ mice per group; $\#p < 0.05$; F $= 4.88$, Fig. 2e−h), thus suggesting a less precise knowledge of the exact position of the platform in the target quadrant. Moreover, after the goal reversal (obtained by displacing the platform to the South quadrant), HFD mice showed delayed learning of the new position of the platform. Indeed, a significant increase in the latency and path length at the fifth day of training was observed ($n = 6−9$ mice per group, latency $\#p < 0.05$; F $= 4.9$; path length $p = 0.05$; F $= 4.3$, Fig. 2b, c). HFD mice also showed reduced goal-related plasticity because of the number of trials needed by an animal to regain the platform position (lean,

$n = 4.1 \pm 0.6$; HFD, $n = 7.4 \pm 1$; $\#p < 0.05$). The observed impairment after the goal reversal is specific for spatial reference memory, as no impairment was observed when they had to shift from the beacon to the spatial strategies (day 1 to days 2−5). Interestingly, the behavioral performance of HFD mice nicely recapitulated that observed following genetic constitutive alteration of adult neurogenesis in mice showing no general impairment in hippocampus-dependent learning, but manifesting a less precise spatial encoding and less flexible updating of the previously learned spatial information[48,49].

**Adult neurogenesis is altered in the DG of obese mice.** Since the impairment in PS could be strictly dependent on the rate of AHN[12], we next examined the neuronal progenitor cells (NPCs) proliferative and neurodifferentiative efficiency by mean of quantitative stereological counting of Ki67 or DCX immunolabeled cell profiles in the DG of lean and obese (HFD and ob/ob) mice. An impairment of the proliferative rate was found in the DG of obese mice compared to respective age-matched lean mice fed on SFD because of the reduction in the number of cells immunoreactive to Ki67, a nuclear protein marker expressed in all phases of the cell cycle except the resting one and usually adopted as an endogenous marker of proliferation in the DG (Fig. 3a) (lean $= 199 \pm 3.6$, HFD $= 170.1 \pm 3.2$, ob/ob $= 159.20 \pm 3.5$; the values are the mean number of immunoreactive neurons/section $\pm$ SEM from $n = 30$ slices from $n = 6$ mice per group; Kruskal−Wallis test and post hoc Dunn's test, $****p < 0.0001$, Kruskal−Wallis statistic $= 35.86$). On the contrary, the number of cells expressing doublecortin (DCX), a marker of neuroblasts and newly immature generated neurons, revealed a significant increment in HFD and ob/ob mice, showing an enhanced rate of neurodifferentiation in obese mice (lean $= 172.17 \pm 2.3$, HFD $= 251.93 \pm 3.4$, ob/ob $= 262.20 \pm 2.7$; the values are the mean number of immunoreactive neurons/section $\pm$ SEM from $n = 30$ slices from $n = 6$ mice per group; ANOVA test with Bonferroni post hoc, $****p < 0.0001$, F $= 291.0$) (Fig. 3b). These data were further supported by a concomitant enhancement, in the DG of obese vs lean mice, of the number of cells immunopositive to NeuroD, a precocious marker of neurodifferentiation expressed before DCX by newly generated immature neurons (lean $= 164 \pm 2.3$, HFD $= 298 \pm 2$, ob/ob $= 295.4 \pm 1.1$; the values are the mean number of immunoreactive neurons/section $\pm$ SEM from $n = 30$ slices from $n = 6$ mice per group; Kruskal−Wallis test and post hoc Dunn's test, $****p < 0.0001$, Kruskal−Wallis statistic $= 60.18$) (Fig. 3c). Several reports suggest that OxA, 2-AG, and leptin may exert a morphoregulatory effect on neuronal structures such as dendritic arborization[39,50,51]. To investigate this aspect in the hippocampus of lean and obese mice we examined the dendritic development, length, and branching complexity of DCX-positive neurons by their morphological and morphometric examination at the inner part of the dentate molecular layer (ML), the region where the synapses of the medial perforant path (MPP) are located. Furthermore, the functional integration of newborn neurons in the DG was estimated by

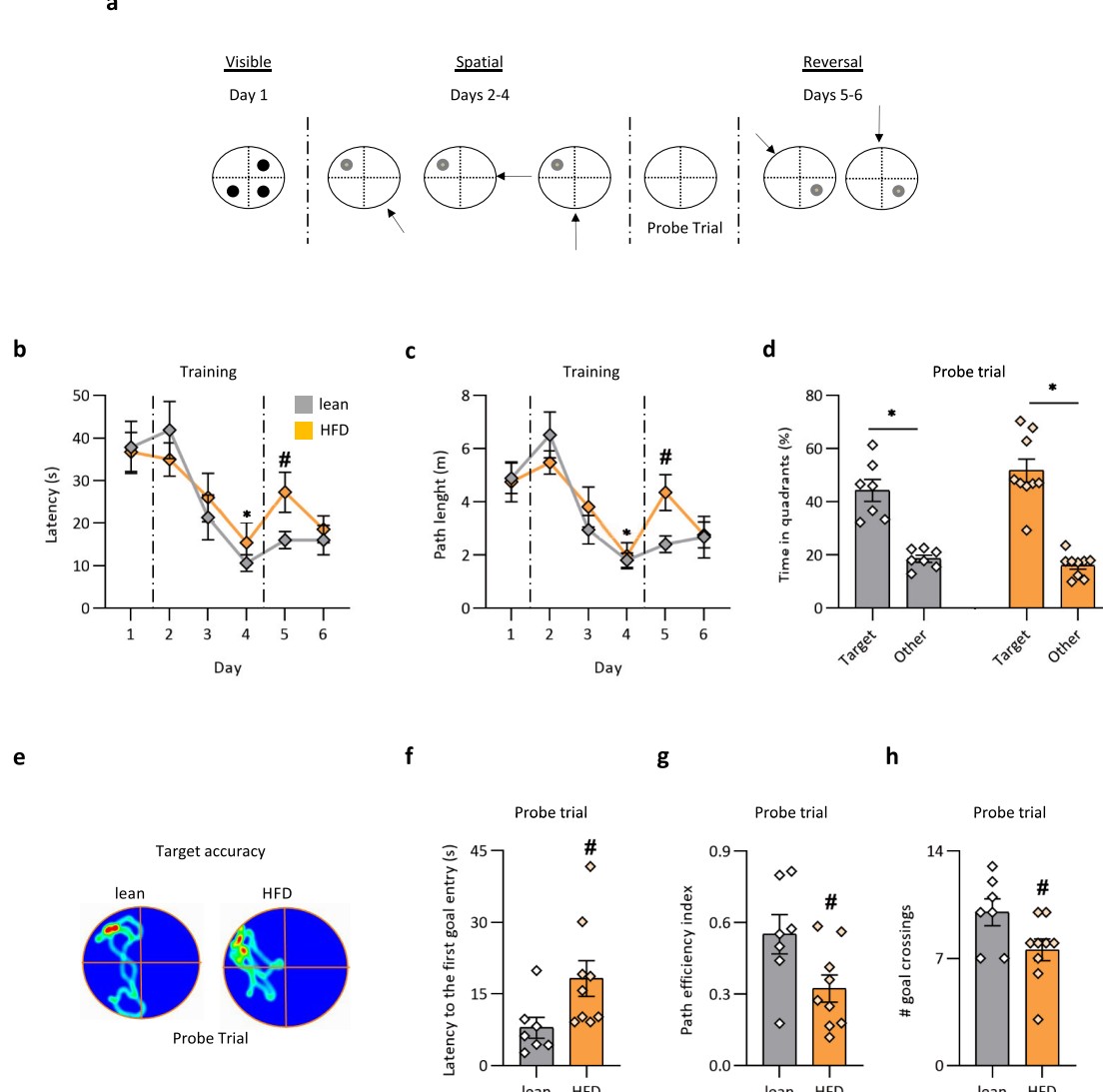

**Fig. 2 Differences in spatial learning abilities of lean and HFD mice in the water maze. a** Schematic representation of the water maze protocol (the black and gray dot inside the pool represents the position of the visible and hidden platform, respectively). **b**, **c** Latency and path lengths to reach the platform during the visible task and training. ANOVA test and Bonferroni post hoc test; latency *$p < 0.0001$; $F = 24.37$; path length *$p < 0.0001$; $F = 22.89$; $n = 7$ lean mice and $n = 9$ HFD mice. **d** Percentage of time spent in the target quadrant compared to the others during the probe trial (lean: target quadrant = 44.29% ± 4.1%, other quadrants = 18.57% ± 1.3%; HFD: target quadrant = 51.72% ± 4.3% and other quadrants = 16.07% ± 1.4%. ANOVA test and Bonferroni post hoc test, *$p < 0.0001$; $F = 56.9$; $n = 7$ lean mice and $n = 9$ HFD mice. **e** Representative target accuracy in the probe trial was indicated by heat maps. Dark red zones represent a sixfold presence probability. **f**–**h** Latency to the first goal entry (lean = 7.9 ± 2.1, HFD = 18.21 ± 3.7; #$p < 0.05$; $F = 4.88$), path efficiency (lean = 0.5 ± 0.08, HFD = 0.3 ± 0.05; #$p < 0.05$; $F = 5.55$) and goal crossings (#$p < 0.05$; $F = 4.82$; lean = 10 ± 0.8, HFD = 7.5 ± 0.7) are all measures related to a precise localization of the platform position in the quadrant during the probe trial. Histograms represent mean ± SEM. *comparison within the groups; #comparison between the groups. $n = 7$ lean mice and $n = 9$ HFD mice. Source data are provided as a Source Data file.

evaluating the density of distribution of PSD95/DCX-positive puncta in lean and obese mice, assessed as a percentage of PSD95/DCX colocalizing-ir area vs DCX-ir area in the inner (IML) and medial (MML) molecular layers of the DG, collectively indicated as ML. We found that such measure was significantly higher in obese mice (lean = 19.53% ± 0.9%; HFD= 34% ± 1.1%; ob/ob = 29.77% ± 1.1%; the values are mean percentage ± SEM of PSD95/DCX-ir area (μm²) vs DCX-ir area (μm²) measured in the IML-MML region of DG from $n = 30$ slices from $n = 6$ mice per group; ANOVA test with Bonferroni post hoc, ****$p < 0.0001$, $F = 47.48$) (Fig. 3d–g). Furthermore, we observed that DCX-positive neurons of obese mice displayed a higher degree of

maturation and complexity of dendritic processes in comparison to those of respective lean mice, as indicated by the branching of DCX-positive secondary (md), primary (pd) and granular cell-derived dendrites (gcld) (Fig. 3h−k) (md: lean = 6.3% ± 0.3%, HFD = 10.6% ± 0.7, ob/ob = 9.66% ± 0.5%; pd: lean = 3.4% ± 0.1%, HFD = 7.4% ± 0.3%, ob/ob = 9.8% ± 0.5%; gcld: lean = 3.2% ± 0.5%, HFD = 3.13% ± 0.6%, ob/ob = 4.5% ± 0.5%; the values are the mean percentage of DCX-ir area±SEM/μm² of each respective GCL, IML and MML layers morphologically identified as depicted in Fig. 3l. Data are from $n = 30$ slices from $n = 6$ mice per group; two-way ANOVA with Tukey post hoc, **$p < 0.01$, ***$p < 0.001$, ****$p < 0.0001$).

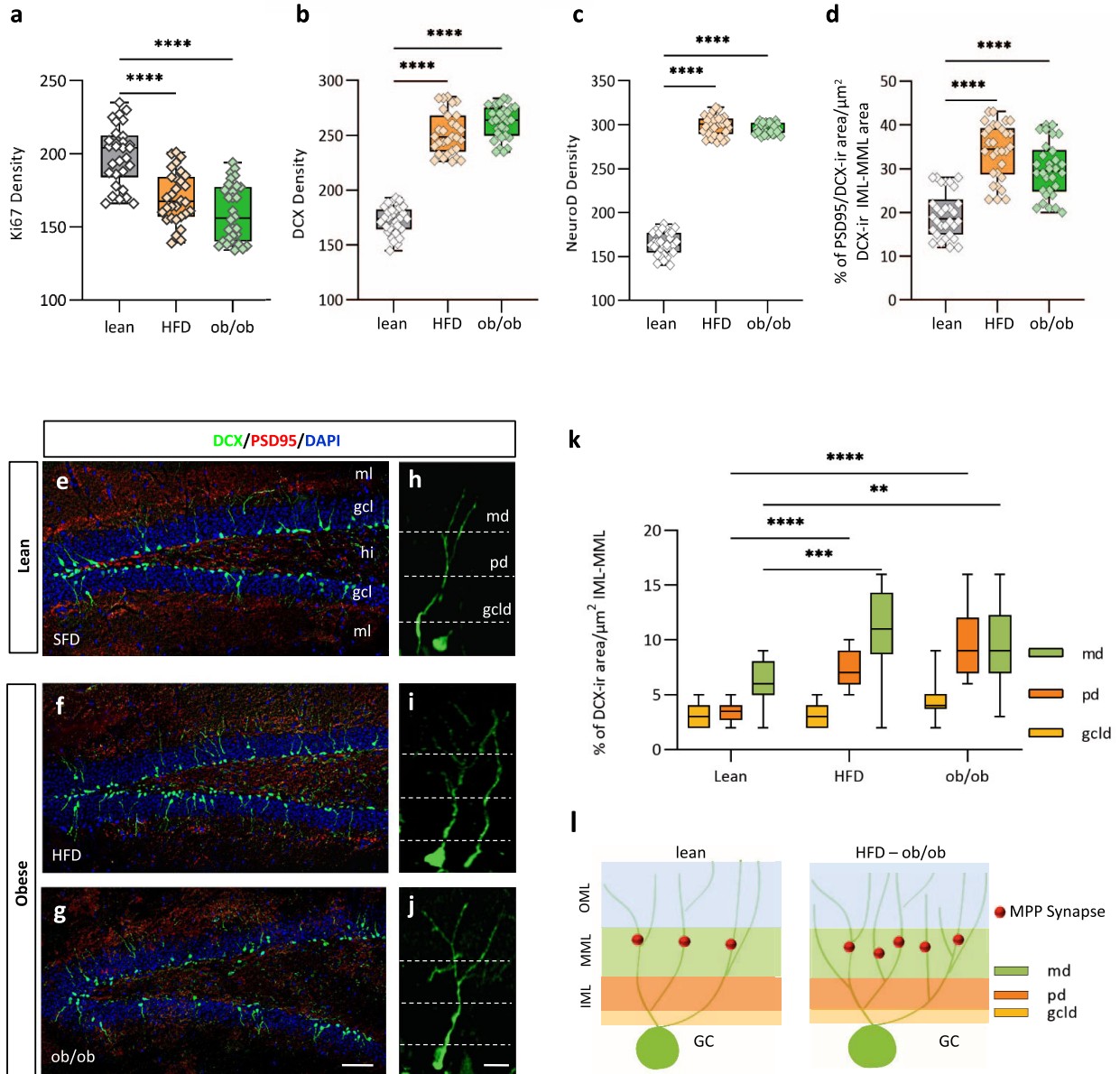

**Fig. 3 Adult hippocampal neurogenesis is altered in the DG of obese mice. a−c** Graphs showing the NPCs proliferative and differentiative efficiency by mean of quantitative stereological counting of Ki67-ir (**a**), or DCX-ir (**b**) or NeuroD-ir (**c**) cell profiles in the DG of lean and obese (ob/ob and HFD) mice. Graphs show box-whisker plots (including minima, maxima, and median values, lower and upper quartiles) with single values. Data are from n = 30 slices in n = 6 mice per group; ANOVA test with Bonferroni post hoc, ****p < 0.0001, F = 47.48). **d** Graph showing dendritic development and complexity of DCX-positive neurons by quantification of PSD95/DCX-ir colocalizing area vs DCX-ir area in the inner (IML) and medial (MML) molecular layers of DG in lean and obese mice. Graphs show box-whisker plots (including minima, maxima, and median values, lower and upper quartiles) with single values. Data are mean of the percentage ± percentage of SEM from n = 30 slices in n = 6 mice per group; ANOVA test with Bonferroni post hoc, ****p < 0.0001, F = 47.48). **e−g** Confocal images of DCX/PSD95/DAPI immunolabeled of the dentate gyrus (DG) showing the adult-born granule cells DCX-ir (green) which constitute a subpopulation of the granular cell layer with higher PSD95-ir (red) postsynaptic density in the IML and MML of obese (**f, g**) and lean (**e**) mice as demonstrated by the high degree of colocalizing DCX/PSD95-ir puncta (yellow-orange) as an index of functional integration of the adult-born granule cells in the DG revealed by the high-density percentage of PSD95/DCX-ir colocalizing area vs DCX-ir area in the ML of obese mice (lean: 19.5% ± 0.9%, HFD: 34.2% ± 1.1%, ob/ob: 29.8% ± 1.1%. n = 30 slices in n = 6 mice per group; ANOVA test with Bonferroni post hoc, ****p < 0.0001, F = 47.48) (scale bar: 100 μm). Immunolabeling was repeated n = 3 times independently in sections from n = 6 mice per group, with similar results. **h−j** High-power images of typical DCX-ir adult-born granule cells demonstrating the development, length, and branching complexity of DCX-positive neurons in obese than in lean mice (scale bar: 10 μm). **k** Percentage of basal, proximal, and medial dendrites revealed by PSD95/DCX-ir vs DCX-ir colocalizing area in the respective GCL, IML, and MML (ML) were also significantly different between DG of obese and lean mice. Two-way ANOVA with Tukey post hoc, **p < 0.01, ***p < 0.001, ****p < 0.0001. Data are mean of the percentage ± percentage of SEM from n = 30 slices in n = 6 mice per group: Immunolabeling was repeated n = 3 times independently in sections from n = 6 mice per group, with similar results. **l** Schematic representation of the cytoarchitectonic organization of the molecular layer (ML) in DG of lean and obese, HFD and ob/ob, mice. GCL: granular cell layer, IML: inner molecular layer, MML: medial molecular layer, GC: granular cell, md: medial dendrites, pd: proximal dendrite, gcld: granular cell layer dendrite. The box plots elements are: center line, median (Q2); square symbol, mean; box limits, 25th (Q1)−75th (Q3) percentiles; whisker length is determined by the outermost data points. Source data are provided as a Source Data file.

**LTP at MPP-DG synapses in obese mice.** Next, we examined whether the changes in AHN observed in obese mice were accompanied by a functional modification in synaptic plasticity. Newborn neurons display a specific form of LTP that can be elicited without $GABA_A$ receptor blockade[15–17]. This form of neuronal plasticity influences episodic memory formation[18,19]. Thus, we used field evoked postsynaptic potential (fEPSP) recording in the ML of the DG and high-frequency stimulation trains of the MPP to compare the AHN-dependent LTP (ACSF-LTP) in acute brain slices of ob/ob and lean mice; the paired-pulse depression was used to distinguish the MPP to the lateral perforant path (LPP) (Fig. 4a). However, when comparing the percentage of LTP no significant differences were observed between lean and ob/ob mice (Fig. 4b–d). To elucidate the possible contribution of newborn neurons to the impairment of PS we analyzed the MPP-DG-mediated LTP in lean, HFD, and ob/ob mice in an in vivo experimental configuration (Fig. 4e–h). In both HFD and ob/ob mice we could not evoke an LTP when theta-burst stimulation (TBS) was applied to the fibers of the MPP-DG synapses (slope 40−60 min HFD post-TBS: 94.01% ± 8.03%, $n = 4$; and ob/ob post-TBS: 103.55% ± 13.48%, $n = 5$), as opposed to the LTP magnitude obtained in lean animals (slope 30−60 min post-TBS: 251.4% ± 16.09%, $n = 5$, Kruskal−Wallis test and post hoc Dunn's test, $*p < 0.05$, Kruskal−Wallis statistic = 9.1) (Fig. 4g, h). These experiments suggest that alterations occurring in the MPP-DG pathway in obese mice, and the subsequent reduction of LTP, may be mediated by endogenous mediators, whose effect during the in vitro LTP recording is lost because of their dilution in the ACSF.

**OxA modulates AHN and MPP-DG plasticity.** To uncover the molecular mechanism that could explain the observed discrepancy between in vitro and in vivo LTP recordings in obese mice, we studied the putative contribution of OxA in the regulation of AHN and plasticity in the hippocampus[38,40,41]. Our previous studies showed that LH orexinergic neurons are disinhibited in HFD and ob/ob mice[42,43]. Thus, we hypothesized that the alteration in MPP-DG plasticity could be due to an increase of OxA release in the DG, which receives orexinergic afferents from the LH. To test this hypothesis, first of all, we quantified OxA levels of hippocampi from lean mice injected with vehicle or OxA (40 μg/kg, i.p., 2 h) in comparison to HFD and ob/ob mice, using an ELISA assay, and found a doubling of OxA levels in obese (HFD and ob/ob) vs lean mice, which was mimicked by OxA injection in lean mice (lean = 4.82 ± 0.1 pmol/mg, lean + OxA = 10.10 ± 0.8 pmol/mg, HFD = 10.57 ± 0.2 pmol/mg, ob/ob = 11 + 0.2 pmol/mg), and reverted by leptin treatment (5 mg/kg, i.p., 2 h) in ob/ob mice, where OxA levels were lowered to become similar to those of lean mice (ob/ob = 11 pmol/mg vs. ob/ob mice injected with leptin = 5.02 ± 0.1 pmol/mg; $n = 7$ mice per group, each sample subjected to triplicate measurements, ANOVA test and Bonferroni post hoc test, $***p < 0.001$, F = 222.1) (Fig. 5a). These results support previous observations that the increased OxA release from the LH to the hippocampus of ob/ob mice is due to lack of tonic leptin signaling in the hypothalamus[42].

Next, we tested the hypothesis that the increased OxA release was responsible for the pro-differentiative effect observed in the DG of obese mice. Lean mice were injected with OxA (280 pmol, icv) to mimic the hypothalamic enhancement of OXA levels in obese mice, alone or together with the Ox1-R antagonist SB334867 (30 mg/kg, i.p. injection 12 h before and after the OxA administration). We found, in lean mice, an increment of the neurodifferentiative rate (Supplementary Fig. 2a−c, g) similar to that observed in untreated HFD and ob/ob mice (Fig. 3c) (NeuroD cells in lean mice: vehicle = 163.1 ± 2.1, OxA = 248.8 ±

1.9, OxA + SB = 162.6 ± 2.2; the values are the mean number of immunoreactive neurons/section ± SEM from $n = 30$ slices from $n = 6$ mice per group; Kruskal−Wallis test and post hoc Dunn's test, $****p < 0.0001$, Kruskal−Wallis statistic = 60.81). On the other hand, and in agreement with previous evidence that CB1 receptor activation by endocannabinoids is downstream of OxA signaling via Ox1-R, in ob/ob mice the number of NeuroD-positive cells was lowered by antagonism not only of OX-1R, but also of CB1, with SB334867 or AM251, respectively (Supplementary Fig. 2d−f, h) (ob/ob = 295.4 ± 1.1, ob/ob + AM251 = 171.6 ± 3.9, ob/ob + SB = 161.3 ± 4.2; the values are the mean number of immunoreactive neurons/section ± SEM from $n = 30$ slices from $n = 6$ mice per group; Kruskal−Wallis test and post hoc Dunn's test, $****p < 0.0001$, Kruskal−Wallis statistic = 60.42).

To confirm that elevated OxA signaling may play a role in the enhanced pro-differentiative effect on immature precursors and rate of NeuroD and DCX cells production, observed above in obese mice, we also performed in vitro experiments using human induced pluripotent stem cell-derived neural stem cells (h-iPSC-derived neural stem cells, NSCs, Axol ab 0013), which are known to express not only Hcrt1R (encoding for the OxA receptor-1 [Ox1-R]) but also Cnr1 (encoding the CB1 receptors), according to the datasheet of the cells Axol Bioscience©. After having confirmed that inhibition of Ox1-R or 2-AG biosynthetic enzyme DAGLα by SB334867 or O-7460 respectively, does not affect the basal differentiation rate of h-iNSCs, the iPSC-derived NSCs were cultured 3 days in presence of OxA to identify the most neurodifferentiative dose, among 50, 100, and 200 nM, using DCX-ir as a marker of neuronal differentiation (h-iNSCs percentage of differentiation: Veh = 9.4% ± 0.55%, SB = 10.3% ± 0.86%, O-7460 = 9.1% ± 0.87%, OxA 50 nM = 55.87% ± 1.7%, OxA 100 nM = 60.73% ± 1.8%, OxA 200 nM = 90.13% ± 0.7%; Supplementary Fig. 3a−f, j). We found that 200 nM was the most effective OxA concentration at promoting both neuronal differentiation and branching of neuronal processes of h-iPSCs. These effects were prevented by adding SB334867 (10 μM) before treatment with OxA 200 nM in the cell medium (h-iNSCs percentage of differentiation: OxA 200 nM + SB = 16.90% ± 1.22%; Supplementary Fig. 3g, j). Most importantly, in agreement with the existence of a OxA/Ox1-R/DAGLα/2-AG cascade, the treatment of h-iPSCs cells with the DAGLα inhibitor, O-7460 (10 μM) before incubation with OxA 200 nM, inhibited the neurodifferentiative effects of the neuropeptide (h-iNSCs percentage of differentiation: OxA 200 nM + O-7460 = 48.37% ± 1.7%; Supplementary Fig. 3h, j). In line with these results, preincubation with SB334867 + O-7460 completely inhibited the OxA 200 nM effect on h-iPSCs differentiation (h-iNSCs percentage of differentiation: OxA 200 nM + SB + O-7460 = 8.7% ± 0.7%, Supplementary Fig. 3i, j; Kruskal−Wallis test and post hoc Dunn's test, $**p < 0.01$, $****p < 0.0001$, Kruskal−Wallis statistic = 224.2; $n = 9$ wells/condition consisting in $n = 3$ biological replicates each with $n = 3$ technical replicates; Supplementary Fig. 3j).

Finally, we tested the hypothesis that the enhancement of OxA levels in the ML-DG (Fig. 5a) could be also the cause of the impairment of AHN-dependent LTP in this area, observed in vivo. Brain slices of lean mice were preincubated with increasing concentrations of OxA among which OxA 200 nM caused an LTP impairment that was restored by antagonizing Ox1-R with SB334867 (10 μM) (slope 40−60 min post HFT: 120.5% ± 6.938% from control slices, $n = 10$ slices from four lean mice; 116.8% ± 6.938% from eight slices treated with 100 nM OxA from four lean mice; 91.83% ± 6.933% from 11 slices treated with 200 nM OxA from five lean mice, slope 40−60 min post HFT: 116.1% ± 7.023%, $n = 11$ slice treated with OxA + SB from five lean mice, ANOVA with Bonferroni test, F = 4.90, $*p < 0.05$) (Fig. 5b−d).

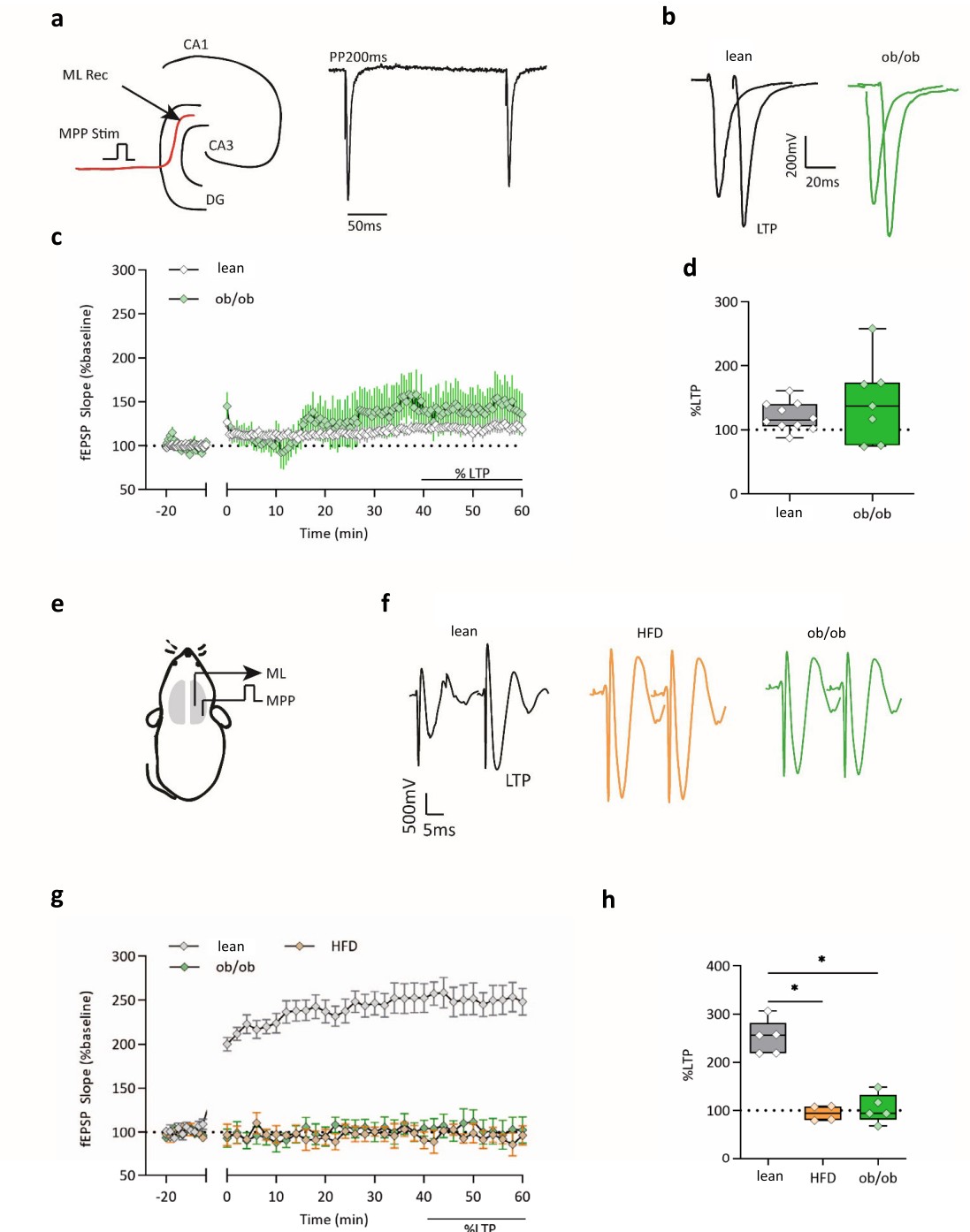

**Fig. 4 Obesity impairs LTP at the medial perforant pathway. a** Experimental configuration of LTP recording in lean and ob/ob mice in brain slices. Position of the stimulating and recording electrodes in the molecular layer (ML Rec) and medial perforant path (MPP stim), respectively. The stimulation of the MPP fibers induces a paired-pulse depression. **b** Representative fEPSPs were recorded before and after the induction of the LTP in the two genotypes. **c** Plot of the fEPSP slope recorded in brain slices before and after the induction of the LTP (time 0) in the DG of lean and ob/ob mice. Data are presented as mean values ± SEM. The extent of LTP was calculated as a percentage of the baseline between the last 40 and 60 min of recording. $n = 10$ slices from four lean mice and $n = 7$ slices from four ob/ob mice. **d** Box plot of the LTP in lean and ob/ob mice, $n = 10$ slices from four mice in lean (slope 40−60 min post HFT = 120.5% ± 6.9%) and seven slices from four ob/ob mice (slope 40−60 min post HFT = 143.5% ± 24.37%). **e** Schematic representation of the in vivo LTP recording, the stimulation electrode was placed in the MPP, the recording site was in the ML. **f** Representative traces recorded in lean, HFD, and ob/ob before and after the TBS. **g** Plot of the fEPSP slope recorded before and after the induction of the LTP (time 0) in the DG of $n = 5$ lean mice, $n = 4$ HFD mice, and $n = 5$ ob/ob mice; data are presented as mean values ± SEM, as the extent of LTP was calculated as a percentage of the baseline between the last 40 and 60 min of recording. **h** Box plot of the LTP in lean, HFD, and ob/ob mice, $n = 5$ lean mice; $n = 4$ HFD and $n = 5$ ob/ob mice. TBS in lean: 251.4% ± 16.09%; 94.01% ± 8.03%, in HFD, 103.55% ± 13.48% in ob/ob, Kruskal−Wallis test and post hoc Dunn's test, *$p < 0.05$, Kruskal−Wallis statistic = 9.1. The box plots elements are: center line, median (Q2); square symbol, mean; box limits, 25th (Q1)−75th (Q3) percentiles; whisker length is determined by the outermost data points. Source data are provided as a Source Data file.

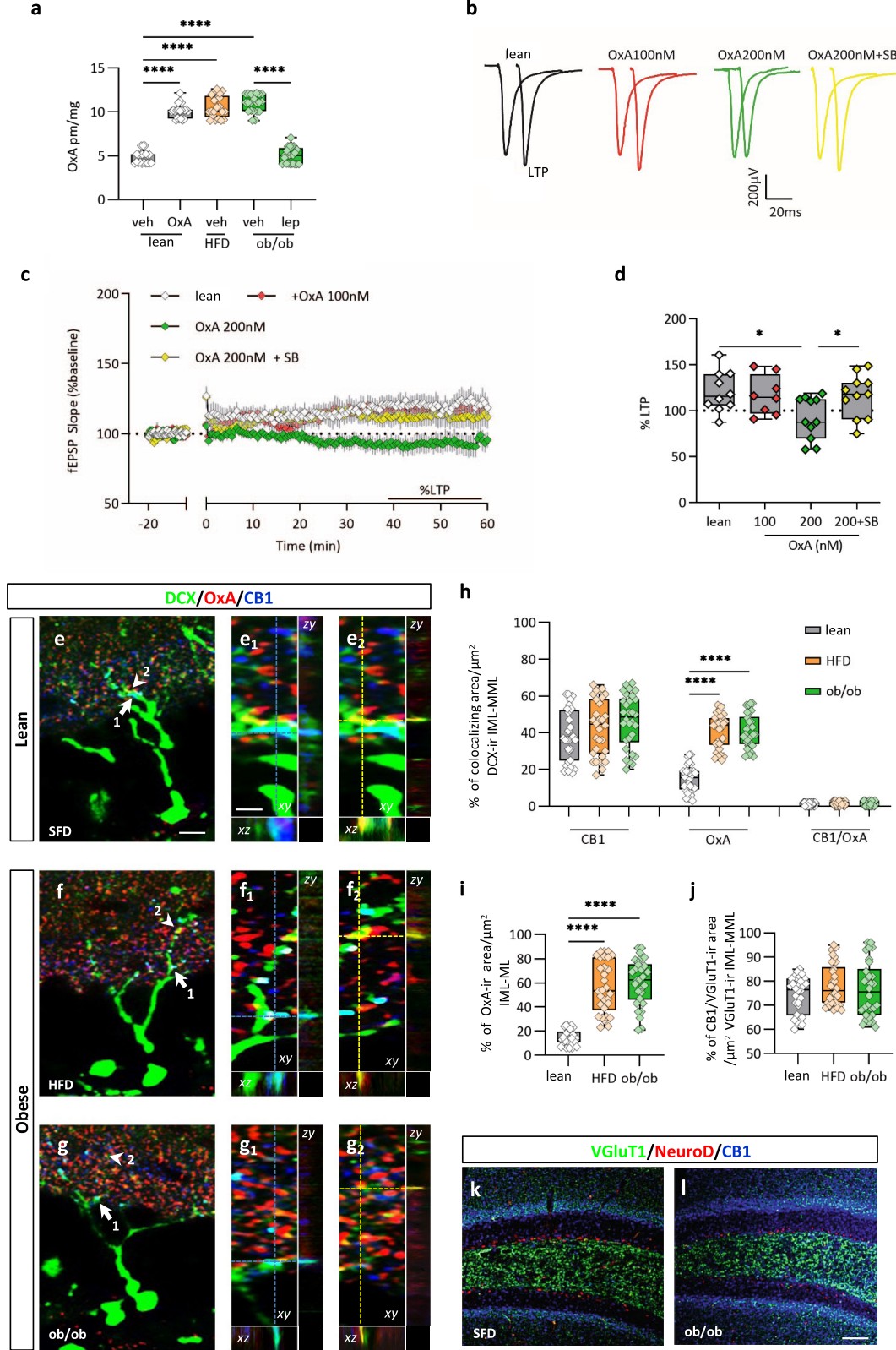

**OxA elicits 2-AG biosynthesis at ML-DG synapses**. As it was previously demonstrated, DG newborn neurons express a functional eCBs system[31–33]. To investigate the putative existence of the anatomical substrate underlying the eCBs and OxA interplay at the ML of DG, we first performed a confocal study of multiple DCX/OxA/CB1 immunolabeling, in lean and obese mice. A wide density of distribution of CB1/DCX-ir puncta was found in the IML-ML of DG (Fig. 5e−g, arrows and Fig. 5e₁−g₁), without differences between lean and obese mice (lean = 38.37% ± 2.5%, HFD = 42.70% ± 2.7%, ob/ob = 46.37% ± 2.6% expressed as mean of the percentage of extension of each co-immunoreactive area/μm² of DCX-ir area ± percentage of SEM from $n = 30$ slices in six mice per group) (Fig. 5h), whereas an increase of OxA/DCX-ir puncta (Fig. 5e−g arrowheads and Fig. 5e₂−g₂) was

**Fig. 5 OxA increases neuronal differentiation and impairs AHN-dependent LTP in brain slices of the hippocampus. a** Box plot of the OxA levels in lean + vehicle (lean), lean + OxA (40 μg/kg, i.p), HFD + vehicle, ob/ob, ob/ob + Lep (5 mg/kg). lean + veh: 4.82 ± 0.1 pmol/mg, lean + OxA: 10.10 ± 0.8 pmol/mg, HFD + vehicle = 10.57 ± 0.2 pm/mg, ob/ob + veh: 11 + 0.2 pmol/mg, ob/ob + leptin 5.02 ± 0.1 pmol/mg. $n = 21$ (7 mice per group in triplicate), ANOVA test and Bonferroni post hoc test, ****$p < 0.0001$, F = 222.1. **b** Representative fEPSPs recorded before and after the induction of the LTP in lean, lean + OxA 100 nM, lean + OxA 200 nM, lean + OxA 200 nM + SB 10 μM. **c** Plot of the fEPSP slope recorded in brain slice before and after the induction of the LTP (time 0) in the DG of lean slices, lean + OxA (100 nM), lean + OxA (200 nM) and lean + OxA + SB (200 nM OxA + SB 10 μM). $n = 10$ slices from four mice (lean), $n = 8$ slices from four mice (lean + OxA 100 nM), $n = 11$ slices from five mice (lean+ 200 nM OxA or lean+ 200 nM OxA+ SB), data are presented as mean values ± SEM. **d** Box plot of the LTP in the four experimental conditions plotted in (**c**). $n = 10$ slices from four mice (lean), slope 40−60 min post HFT: 120.5% ± 6.938% from control slices, $n = 8$ slices from four mice (lean + OxA 100 nM), slope 40−60 min post HFT: 116.8% ± 6.938%, $n = 11$ slices from five mice for lean+ 200 nM OxA and lean+ 200 nM OxA +SB groups, slope 40−60 min post HFT: 91.83% ± 6.933% and 116.1% ± 7.023% respectively. *$p < 0.05$; ANOVA with Bonferroni post hoc test, F = 4.90. **e**−**g** Confocal images of DCX/OxA/CB1 immunolabeling of dentate gyrus (DG) showing a wide density of distribution of CB1/DCX-ir (light blue, arrow) or OxA/DCX-ir (yellow-orange, arrowhead) colocalizing puncta in the IML-MML of lean (**e**) and obese (**f**, **g**) mice. **e₁−g₂** High-power fluorescent micrographs of orthogonal stacks are shown for each respective (**e**−**g**) area; dotted lines and crosshairs are used to show 3D coordinates and define the area of interest indicated by arrows 1 and arrowheads 2 which correspond to numbered insets in (**e**−**g**). Scale bar: 10 μm (**e**−**g**) and 2 μm (**e₁**−**g₂**). **h**−**j** Graphs showing the percentage of CB1/DCX-ir or OxA/DCX-ir or CB1/OxA/DCX-ir colocalizing area/μm² of DCX-ir area in the IML-MML (**h**), or OxA-ir area/μm² of IML-MML area (**i**), or CB1/VGluT1 colocalizing area/μm² of VGluT1-ir area in the IML-MML (**j**), in lean and obese mice. Graphs show box-whisker plots (including minima, maxima, and median values, lower and upper quartiles) with single values. Data are mean of the percentage ± percentage of SEM from $n = 30$ slices in six mice per group; Kruskal−Wallis test and post hoc Dunn's test, ****$p < 0.0001$). **k**, **l** Confocal images of VGluT1/NeuroD/CB1 immunolabeling of DG in lean (**k**) and obese ob/ob mice (**l**) showing a wide density of distribution of CB1/VGluT1 (light blue) colocalizing puncta in the IML-MML. Scale bar: 200 μm. Immunolabeling was repeated $n = 3$ times, independently in different DG sections from $n = 6$ mice per group, with similar results. The box plots elements are: center line, median (Q2); square symbol, mean; box limits, 25th (Q1)−75th (Q3) percentiles; whisker length is determined by the outermost data points. Source data are provided as a Source Data file.

---

observed in obese mice (lean = 14.63% ± 1.1%, HFD = 30.50% ± 3.8%, ob/ob = 34.6% ± 5.4%, as mean percentage density of colocalization/μm² of DCX-ir IML-MML area ± percentage of SEM from $n = 30$ slices in six mice per group; Kruskal−Wallis test and post hoc Dunn's test, ****$p < 0.0001$, Kruskal−Wallis statistic = 57.5) (Fig. 5h). The lack of CB1/OxA/DCX-ir triple colocalizing puncta (lean = 0.1% ± 0.008%, HFD = 0.08% ± 0.005%, ob/ob = 0.15% ± 0.003% as mean percentage density of colocalization/μm² of DCX-ir IML-MML area ± percentage of SEM from $n = 30$ slices in six mice per group) (Fig. 5h) revealed in the IML-ML DG area the presence of two different clusters of CB1 and OxA afferences to DCX-ir dendritic spines belonging to different pathways, of which the orexinergic ones were coming from the lateral hypothalamus. In agreement with the biochemical data (Fig. 5a), an elevation of OxA immunodensity was revealed in the IML-MML DG of obese mice (lean = 14.63% ± 2.1%, HFD = 56.50% ± 6.8%, ob/ob = 60.57% ± 6.4% as mean percentage density of colocalization ± percentage of SEM from $n = 30$ slices from six mice each, Kruskal−Wallis test and post hoc Dunn's test, ****$p < 0.0001$, Kruskal−Wallis statistic = 56.95) (Fig. 5i). In addition, a wide density of CB1/VGluT1 colocalizing puncta was found in the IML-MML DG, without differences between lean and obese mice (lean = 74.17% ± 4.1%; HFD= 78.40% ± 5.5%; ob/ob = 76.43% ± 4.2% as mean percentage of CB1/VGluT1 co-immunoreactive area/μm² of VGluT1-ir area ± percentage of SEM from $n = 30$ slices six mice per group) (Fig. 5j−l).

To test if the increased orexinergic drive to the IML-MML DG of obese mice was able to dampen the AHN-mediated LTP in obese mice via endocannabinoid signaling, the levels of 2-AG were quantified by LC-MS of DG from obese (HFD and ob/ob) mice, lean mice injected with OxA (40 μg/kg, i.p., 2 h), alone or in combination with SB334867 (30 mg/kg, i.p., 3 h alone or 1 h before OX-A injection), and SB334867-injected obese mice (60 mg/kg, i.p., 3 h alone or 1 h before OxA injection). In line with previous works[37,52], we found an enhancement of 2-AG levels in the DG of lean OxA-injected mice that was prevented by SB334867 pre-administration (Fig. 6a), which, per se, lowered 2-AG levels in the hippocampus of both HFD and ob/ob mice (lean = 3.2 ± 0.5 pmol/mg, lean injected with OxA = 14.1 ± 1.4 pmol/mg lean injected with OxA+SB = 4.3 ± 0.5 pmol/mg,

HFD = 19.9 ± 0.6 pmol/mg, HFD injected with SB = 7.8 ± 0.6 pmol/mg, ob/ob = 18.9 ± 0.8 pmol/mg, ob/ob injected with SB = 8.1 ± 0.3 pmol/mg; $n = 8$ mice per group; ANOVA test and Bonferroni post hoc, ****$p < 0.0001$, F = 84.35) (Fig. 6a). These data further confirm in the DG the direct role of OxA in promoting 2-AG biosynthesis via Ox-1R activation, as already demonstrated in other brain regions[53].

To investigate the anatomical substrate underlying eCB and OxA interplay at ML-DG circuitries, we performed a DCX/PSD95/CB1 confocal microscopy investigation (Fig. 6b−d). In addition to confirming their high degree of synaptogenesis in the ML-DG area in obese mice (Fig. 3), we found the association of DCX-ir cell processes to CB1-positive inputs, as demonstrated by the wide density of distribution of CB1/PSD95/DCX-ir puncta in obese in comparison to lean mice (lean = 36.5% ± 5.7%, HFD = 35.50% ± 6.9%, ob/ob = 43.37% ± 6.6%; the values are mean of the percentage of DCX/PSD95/CB1-immunoreactive area/μm² of DG area ± percentage of SEM from $n = 30$ slices in six mice per group) (Fig. 6b−d and 6b₁−d₁). To reveal conclusively the ultrastructural details of eCB and OxA networks in the ML-DG area, which contains the MPP-ML synapses, we performed a correlative light and electron microscopy (CLEM) study of OX-1R and CB1 immunolabeling in the ML-DG area of POMC-eGFP mice, which represent a widely recognized model of ectopic labeling of DCX-like newborn neurons of the DG[54]. CLEM analysis revealed the dendritic spines of the newborn-eGFP neurons expressing OX-1R labeling (10 nm immunogold particles) at postsynaptic sites of asymmetrical (i.e. putative excitatory) endings containing OxA dense-core vesicles (20 nm immunogold particles) (Fig. 6e). Dendritic spines of newborn-eGFP neurons, carrying DAGLα at their postsynaptic side (10 nm immunogold particles) were found as opposed to OxA-positive puncta (20 nm immunogold particles) at asymmetrical (i.e. putative excitatory) synapses (Fig. 6f). In the same area, the double DAGLα/Ox-1R or DAGLα/CB1 immunogold labeling revealed the expression of DAGLα, the main biosynthesizing enzyme of 2-AG, in the close proximity of Ox-1R-expressing dendritic spines (Fig. 6g), or spines widely contacted by presynaptic nerve endings carrying CB1 receptors (Fig. 6h) at asymmetrical, i.e. putative excitatory, synaptic clefts.

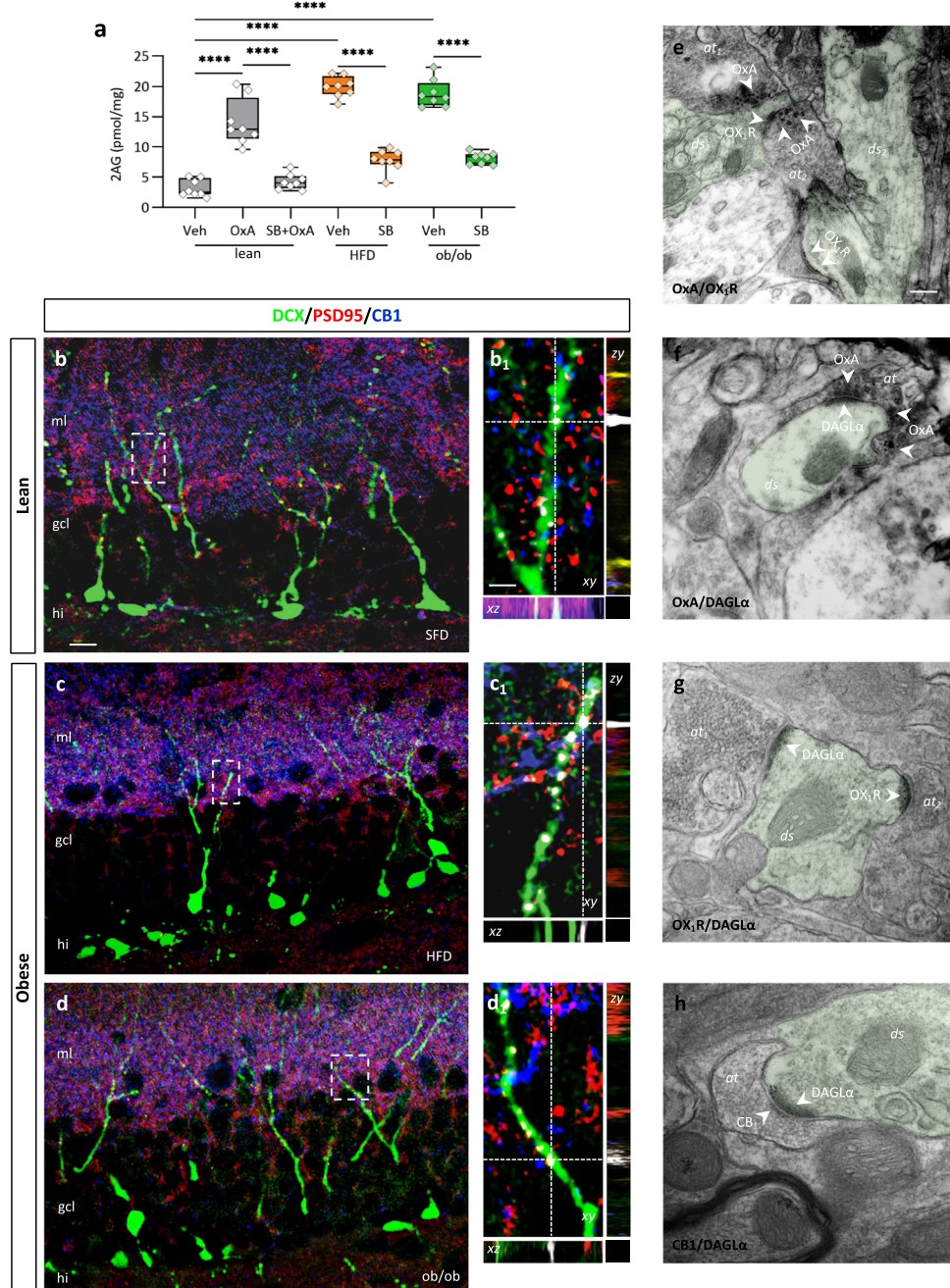

Based on the results described in this section, we hypothesized that in obese mice: (1) elevated OxA release from LH afferents elicits postsynaptic Ox1-R-mediated biosynthesis and release of 2-AG via activation of DAGLα at the postsynaptic dendritic spines of newborn neurons in the ML-DG area, and (2) 2-AG then acts retrogradely at the CB1-positive MPP excitatory inputs to newborn neurons to inhibit glutamate release and action thereupon, with consequent LTP impairment.

**The OxA/eCB system modulates synaptic plasticity at MPP-DG synapses**. To substantiate the above hypothesis, we measured the effect of OxA on MPP-DG LTP following interference with 2-AG/CB1 signaling pathway using CB1 receptor antagonism with AM251 (4 µM) (slope 40−60 min post HFT: 135.8% ± 14.73%, $n = 8$ slices from five lean mice), or DAGLα inhibition with O-7460 (10 µM) (slope 40−60 min post HFT: 121.1% ± 11.82%, $n = 7$ slices from five lean mice). We found that both agents were

able to prevent the inhibitory effect of OxA on LTP in hippocampal slices from lean mice. In line with these results, pre-incubation of hippocampal slices from lean mice with ACEA (125 nM), a synthetic CB1 receptor agonist, also inhibited neurogenesis-dependent LTP (Fig. 7a, b) (slope 40−60 min post HFT: 84.72% ± 6.05%, $n = 9$ slices from five lean mice, Kruskal−Wallis test and post hoc Dunn's multiple comparison test, *$p < 0.05$, Kruskal−Wallis statistic = 11.69). Overall, these data demonstrate that OxA can inhibit this form of LTP via the 2-AG/CB1 signaling system.

Next, we investigated the in vivo effect of Ox1-R or CB1 antagonism with SB334867 (SB—0.1 nmol/0.5 µl) or AM251 (0.5 nmol/0.5 µl), respectively, on LTP by intracranial microinjection in the MPP in obese mice. Microinjection of SB334867 was able to restore LTP in HFD mice, whereas AM251 only produced a small and non-statistically significant counteraction (94.01% ± 8.03% of the baseline in HFD, 195.7% ± 30.39% of the

**Fig. 6 A functional OxA/eCB system is expressed at MPP-DG synapses. a** Box plot of 2-AG concentrations in lean + vehicle (lean), ob/ob, lean + OxA (40 µg/kg, i.p.), lean + OxA (40 µg/kg, i.p) + SB (30 mg/kg, i.p.). $n = 9$ lean mice, $n = 10$ ob/ob mice, $n = 10$ lean injected with OxA (40 µg/kg, i.p.), $n = 10$ lean mice injected with OxA (40 µg/kg, i.p.) + SB (30 mg/kg, i.p), ANOVA with Bonferroni test, F = 84.35; ****$p < 0.0001$. The box plots elements are: center line, median (Q2); square symbol, mean; box limits, 25th (Q1)−75th (Q3) percentiles; whisker length is determined by the outermost data points. **b−d** Confocal images of DCX/PSD95/CB1 immunolabeling of DG in lean and obese mice showing a wide density of distribution of CB1 at PSD95-ir postsynaptic side of DCX-ir dendritic spines. **$b_1$−$d_1$** High-power fluorescent micrographs of orthogonal stacks are shown for each area and referred to the box of the respective (**b−d**) images; dotted with lines and crosshairs are used to show 3D coordinates. Scale bars: 10 µm (**b−d**) and 2 µm (**$b_1$−$d_1$**). Immunolabeling was repeated $n = 3$ times independently in different DG sections from $n = 6$ mice per group, with similar results. **e−h** Correlative light and electron microscopy (CLEM) showing OxA/Ox1-R or OxA/DAGLα or Ox1-R/DAGLα or CB1/DAGLα immunogold reactivity at synapses of immature granule-eGFP cells in the MML of DG. **e** Asymmetrical, putative excitatory, axodendritic synapse between OxA-positive axon terminal (at1) opposite to the dendritic spine (ds1) of immature granule-eGFP cell showing marked Ox1-R immunogold labeling at the edges of the postsynaptic density (arrows). The CLEM analysis revealed the dendritic spines of the newborn-eGFP neurons expressing OX-1R labeling (10 nm immunogold particles at postsynaptic sites of asymmetrical (i.e. putative excitatory)) endings containing OxA dense-core vesicles (20 nm immunogold particles). **f** Dendritic spines of newborn-eGFP neurons, carrying DAGLα enzyme at the postsynaptic side (10 nm immunogold particles) were found as opposed to OxA-positive puncta (20 nm immunogold particles) at asymmetrical (i.e. putative excitatory) synapses. DAGLα/Ox1-R (**g**) or DAGLα/CB1 (**h**) immunogold labeling reveals the expression of DAGLα in the proximity to the Ox1-R dendritic spines (**g**) or the presynaptic nerve endings carrying CB1 receptors (**h**) at asymmetrical, i.e. putative excitatory, synaptic cleft. Scale bar 0.1 µm (**e−h**). Immunolabeling was repeated with similar results $n = 3$ times independently in $n = 6$ different ultrathin sections/mouse from $n = 3$ mice. Source data are provided as a Source Data file.

baseline in HFD + SB; 109.4% ± 4.7% of the baseline in HFD + AM251; Kruskal−Wallis test and post hoc Dunn's multiple comparison test, *$p < 0.05$, Kruskal−Wallis statistic = 8, $n = 4$ mice per group) (Fig. 7c, d). In ob/ob mice, instead, both SB334867 and AM251 microinjection in the MPP induced an elevation of LTP, which was respectively 175.6% ± 3.77% and 190% ± 21.5% vs. the baseline of ob/ob mice in the control condition ($n = 5$ ob/ob, $n = 4$ ob/ob + SB, $n = 4$ ob/ob + AM251 Kruskal−Wallis test and post hoc Dunn's multiple comparison test, *$p < 0.05$, Kruskal−Wallis statistic = 8.69) (Fig. 7e, f). These results support the hypothesis that inhibition of MPP-DG synaptic plasticity in obese mice is associated with an alteration of the orexinergic-eCB pathway. Indeed, the SB334867-mediated enhancement of the excitatory drive at the MPP-DG synapses contributes to the LTP recovery in both HFD and ob/ob mice, at least in part by preventing OxA-mediated 2-AG biosynthesis and release, and the consequent CB1-dependent inhibition of glutamate release from MPP inputs (Fig. 7c−f).

Noteworthy, Ox1R antagonism rescued the LTP impairment of ob/ob and HFD mice, while AM251 did so in ob/ob, but not significantly in HFD mice (Fig. 7c−f), possibly due to the higher 2-AG levels found in the DG of the latter mice. Thus, we performed a further experiment to preliminary investigate the reason for this latter phenomenon. AHN-mediated excitability requires the NMDA receptor, which triggers 2-AG biosynthesis[55–57] and whose GluNR2B subunit is critical to adult-born granule cell synaptic strength because of its lower rate of deactivation than GluNR2A-containing NMDARs[58]. Of note, both expression levels and functioning of GluNR2B are oppositely regulated by leptin and orexin. Electrophysiological studies suggest that one of the mechanisms by which leptin affects memory and synaptic plasticity in the hippocampus is by increasing the levels of surface localized NR2B-containing NMDARs[59–61], while, conversely, OxA may attenuate the induction of the NMDA-dependent form of neural plasticity by inhibiting the NR2B subunit[62,63]. By immunohistochemistry and western blotting analysis of NR2B-ir in the DG of lean and obese mice, we found a robust reduction of NR2B expression (about 80%; $n = 3$) in ob/ob vs. lean mice, whereas this effect was much lower and non-statistically significant in HFD vs. lean mice (Fig. 7g−k). Thus, the lack of significant effect of AM251 on LTP in vivo in HFD mice may be explained with a potential contribution to 2-AG biosynthesis/release of NMDAR activation, which, being the NR2B subunit of this receptor still well expressed in these mice, might explain their slightly higher DG

levels of 2-AG vs. those of ob/ob mice, and hence the lower efficacy of the same dose of AM251 at counteracting their action at CB1. Notably, the reduction of NR2B expression in ob/ob mice was prevented by leptin, which had no effect in HFD mice (Fig. 7j, k). As leptin desensitization occurs mostly, in not uniquely, in the hypothalamic regions of HFD mice, we speculate that there is a link between lack of leptin signaling and functional alterations of NMDAR as proposed previously by other authors[61].

**Inhibition of OxA signaling restores PS in obese mice.** Finally, we inhibited the OxA system to evaluate whether the altered PS could be rescued in HFD and ob/ob mice. We used the NOR as it allows single drug injection. In detail, we observed that a single injection of SB334867 (30 mg/kg for wt or 60 mg/kg for obese mice) 30 min before the acquisition phase in the NOR[64] task did not change the exploratory activity of the identical object in both groups of mice (Fig. 8a, b), which showed a recognition index of 0.12 ± 0.02 for lean, 0.08 ± 0.01 for HFD ($n = 10$) and 0.11 ± 0.02 for ob/ob mice ($n = 10$ mice per group) (see Fig. 1a−d). However, Ox-1R antagonism led to an increase in the recognition index of similar objects when the mice were tested at 1.5-h delay up to 0.36 ± 0.07 in HFD and 0.42 ± 0.06, in ob/ob mice ($n = 10$ per group). Indeed, SB334867 injection reverted the recognition memory deficit at 1.5 h (Fig. 8c), but not at 24-h delay (0.02 ± 0.05 and −0.04 ± 0.06 respectively, ANOVA test and Bonferroni post hoc, **$p < 0.01$, F = 8.6) (Fig. 8d) as reported in Fig. 1d. These results suggest the involvement of OxA signaling in the impairment of the early, but not late, mechanisms of hippocampal processing of episodic memory, observed here in obese mice.

## Discussion

Several studies in humans highlight how a "Western"-style diet, rich in sugars and fats, promotes behavioral deficits and memory impairment associated with obesity[9]. It is widely accepted that episodic memory plays a significant role in satiety, since its impairment reduces the ability for memory retention and planning for eating, and enhances food intake[9]. PS is a process that minimizes the overlap between patterns of neuronal activity representing similar experiences. Recent works suggest that the DG performs this role in memory processing by reducing the similarity of input patterns, an ability that is necessary to discern between similar past events[65]. AHN, a key regulatory factor for PS[11,12], is altered in terms of proliferation and differentiation in

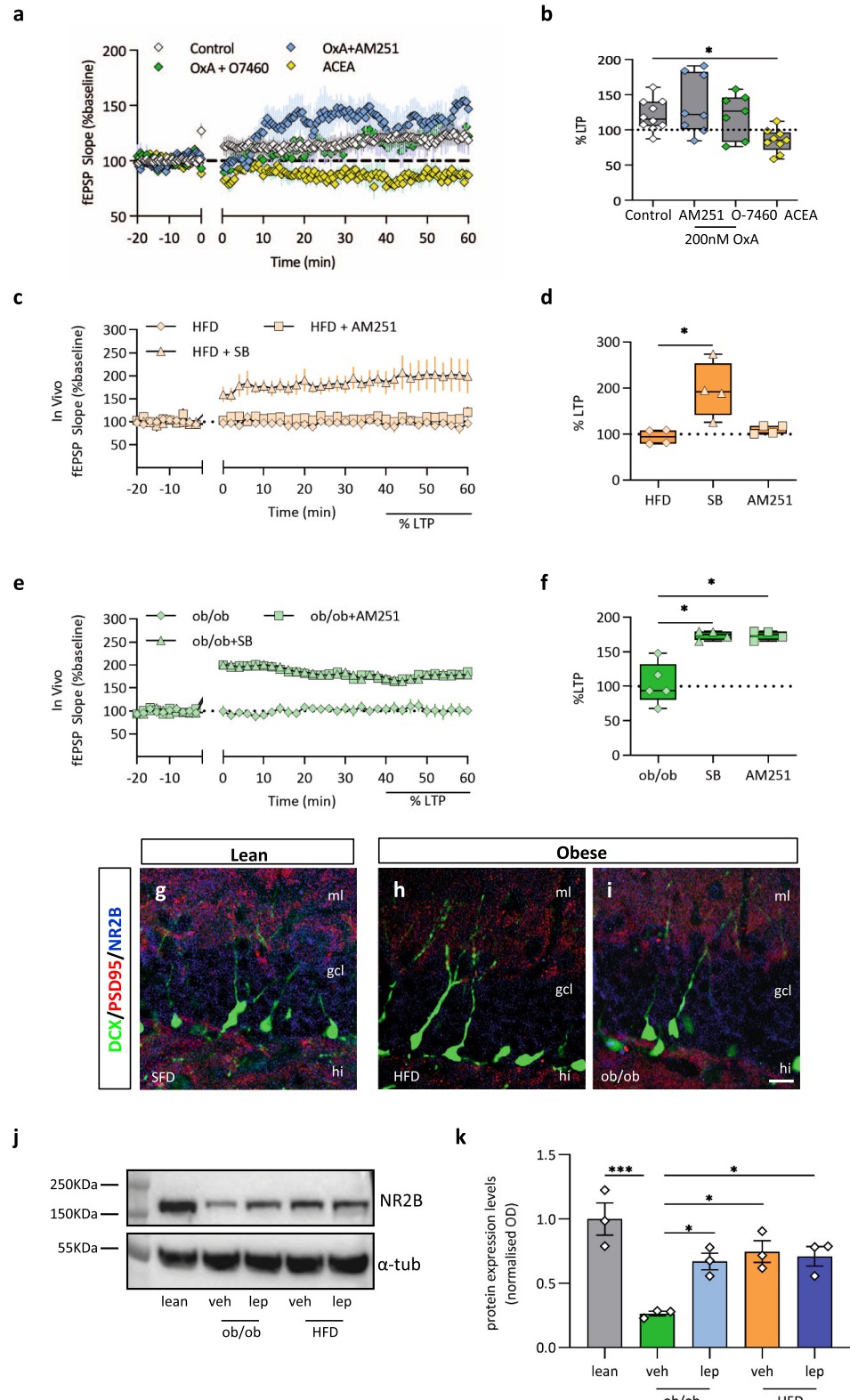

mice made obese by an HFD[24,25] and is functionally associated with impairment in episodic memory[9]. Our data provide a mechanistic molecular framework in which OxA, a hypothalamic neuropeptide over-released during overweight and obesity[42,43], accelerates the onset of differentiation of immature GCs from the subgranular zone toward a neuronal fate, thus altering the synaptic plasticity of MPP-DG synapses via 2-AG-mediated CB1

receptor overactivation. Altogether, the present data suggest that the plasticity of MPP-DG synapses is regulated by 2-AG, which acts retrogradely at CB1 receptors expressed at presynaptic MPP inputs to the dendritic spines of GCs by inhibiting glutamate release. We propose that, as 2-AG overproduction occurs in the DG of obese mice—an effect mainly attributable to Ox-1R overactivation by OxA—and the subsequent aberrant OxA and

**Fig. 7 OxA-mediated impairment of AHN and plasticity of MPP-DG occurs via 2-AG/CB1 signaling. a** Plot of the fEPSP slope recorded in brain slice of lean mice before and after the induction of the LTP (time 0) in lean mice, OxA 200 nM +AM251, OxA 200 nM +O7460, ACEA. $n = 10$ slices (control), $n = 8$ slices (OxA + AM251), $n = 7$ slices (OxA +O7460), $n = 9$ slices (ACEA) from $n = 5$ mice/treatment; data are presented as mean values ± SEM. **b** Box plot of the LTP in the four experimental conditions. $n = 10$ slices control, $n = 8$ from OxA +AM251, $n = 7$ OxA +O7460, $n = 9$ ACEA from five mice each. Slope 40−60 min post HFT in lean mice = 120.5% ± 6.9%; AM251 (4 μM) 40−60 min post HFT: 135.8% ± 14.83%; O-7460 (10 μM) slope 40−60 min post HFT: 121.1% ± 11.82%; ACEA (125 nM) slope 40−60 min post HFT: 84.73% ± 5.5%. Kruskal−Wallis test and post hoc Dunn's multiple comparison test, *$p < 0.05$, Kruskal−Wallis statistic = 11.69. **c** Plot of the fEPSP slope recorded in vivo before and after the induction of the LTP in HFD mice (time 0), $n = 4$ mice per group, data are presented as mean values ± SEM. **d** Box plot of the LTP in the three experimental conditions in HFD mice, HFD + SB (SB—0.1 nmol/0.5 μl) and HFD + AM251 (0.5 nmol/0.5 μl). 195.7% ± 30.39% of the LTP baseline for HFD+SB; 109.4% ± 4.7% of the LTP baseline in HFD + AM251; Kruskal−Wallis test and post hoc Dunn's multiple comparison test, *$p < 0.05$, Kruskal−Wallis statistic = 8, $n = 4$ mice per group. **e** Plot of the fEPSP slope recorded in vivo before and after the induction of the LTP (time 0) in ob/ob mice. $n = 5$ vehicle-injected, $n = 4$ SB-injected and $n = 4$ AM251-injected ob/ob mice; data are presented as mean values ± SEM. **f** Box plot of the LTP in the three experimental conditions in ob/ob mice, ob/ob + SB (SB—0.1 nmol/0.5 μl) and ob/ob + AM251 (0.5 nmol/0.5 μl). LTP was 175.6% ± 3.77% in ob/ob + SB and 190% ± 21.5% in ob/ob + AM251. $n = 5$ ob/ob, $n = 4$ ob/ob + SB, $n = 4$ ob/ob + AM251 Kruskal−Wallis test and post hoc Dunn's multiple comparison test, *$p < 0.05$, Kruskal−Wallis statistic = 8.69. The box plots elements are: center line, median (Q2); square symbol, mean; box limits, 25th (Q1)−75th (Q3) percentiles; whisker length is determined by the outermost data points. **g−i** NR2B downregulation in DG of obese mice. Confocal images of DCX/PSD95/NR2B immunolabeling of DG in lean (**g**), obese HFD (**h**), and ob/ob (**i**) mice showing a wide density of distribution of NR2B-ir (blue) at IML-MML DG of lean mice and colocalization with PSD95-ir (red) at postsynaptic side of DCX-ir dendritic spines. Scale bar: 10 μm. Immunolabeling was repeated $n = 3$ times independently in different DG sections from $n = 6$ mice per group, with similar results. **j, k** Measurement of NR2B subunit expression. Effect of leptin on NR2B subunit expression in lean, ob/ob, and HFD mice treated with or without leptin evaluated by western blot analysis. **j** Representative blot showing the band intensity of NR2B subunit in the aforementioned experimental mice. **k** Bar graphs showing the quantification of phospho-NR2B subunit normalized to α-tubulin. Data represent the means ± SEM from $n = 3$ mice/group. Data sets were compared by one-way ANOVA followed by Tukey's test, *$p < 0.05$, ***$p < 0.001$, F = 10.63. Source data are provided as a Source Data file.

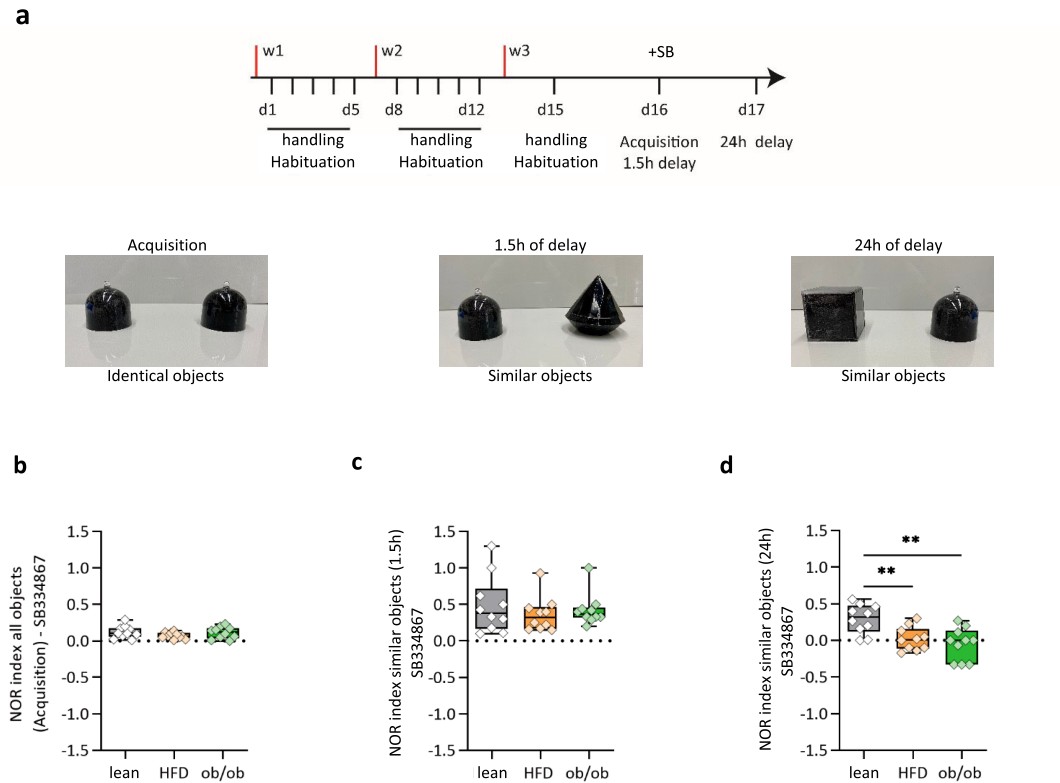

**Fig. 8 The alterations in pattern separation in obese mice are partly rescued by blocking OxA/Ox1-R signaling. a** Representative scheme and timeline of the novel object recognition test, animals were injected i.p. 1.5 h before with SB (30 mg/kg, i.p.). **b** NOR in lean, HFD, and ob/ob mice calculated during habituation phase. **c** Short-term retention phase, NOR in lean, HFD, and ob/ob mice calculated after 1.5 h. **d** Long-term retention phase, NOR = 0.29 ± 0.06 in lean, NOR = 0.02 ± 0.05 in HFD, NOR = −0.04 ± 0.06 in ob/ob mice calculated after 24 h. **$p < 0.01$, ANOVA test and Bonferroni post hoc test, F = 8.6, $n = 10$ mice per group. The box plots elements are: center line, median (Q2); square symbol, mean; box limits, 25th (Q1)−75th (Q3) percentiles; whisker length is determined by the outermost data points. Source data are provided as a Source Data file.

CB1-dependent neurodifferentiative drive and branching of DCX-ir cells, together with enhanced CB1-mediated inhibition of LTP at MPP inputs to the DG, results in impaired PS processing. The present study reveals in obese ob/ob and HFD mice, that overactivity of orexinergic/endocannabinoid signaling at MPP-DG synapses leads to inhibition of AHN-mediated LTP in the DG with detrimental effects on PS and its functional impact on episodic memory. Since this sequence of events may appear to contrast with other studies showing a linear correlation between memory ability and neurogenesis[13,14], we here specifically studied the phenomenon of neurogenesis-dependent LTP[15,18] to unravel this discrepancy in a model of aberrant AHN such as that provided by obese ob/ob and HFD mice. These effects have not been investigated before neither in ob/ob nor HFD mouse models of obesity nor in obese db/db leptin receptor knockout mice, which exhibit AHN alterations similar to those of ob/ob mice[24] despite their reduction of OxA levels in the lateral hypothalamus[66,67].

In both, ob/ob and HFD obese mice, we found mostly identical behavioral, electrophysiological, biochemical, and pharmacological responses, suggesting that the phenomenon reported here may be common to genetic and diet-induced obesity, as well as to murine and human obesity, where an overproduction of OxA and 2-AG is also observed, at least at the peripheral level[53,68,69]. Indeed, the only unexpected difference found here between ob/ob and HFD mice, was the lack of effect of CB1 blockade by AM251 on LTP in vivo in the latter mice, whereas Ox-1R receptor blockade was effective in both groups of obese mice (and CB1 antagonism and agonism blocked and mimicked, respectively, the effect of OxA on LTP in ex vivo lean brain slices). This difference in the actions of AM251 may be due to: (1) 2-AG acting via CB1 downstream of Ox1-R, and hence the effect being more sensitive to Ox-1R than CB1 blockade; and (2) hippocampal 2-AG levels in HFD mice being on average ~1 pmol/mg tissue (i.e. ~1 μM) higher (although not significantly) than in ob/ob mice (Fig. 6a), despite the slightly lower number of new neuroblasts (Fig. 3b), which may have yielded a higher local eCB tone in HFD mice, thus possibly requiring higher concentrations of AM251 to be effectively antagonized.

Indeed, AHN-mediated excitability requires the contribution from NR2B-containing NMDAR receptors[59–61] whose expression is known to be stimulated by leptin and reduced by OxA. We found here that NR2B expression in the DG is significantly reduced in ob/ob but not HFD mice. As leptin receptor desensitization typical of HFD obese mice occurs mostly, if not uniquely, in the hypothalamic regions of HFD mice, whereas excessive OxA levels were found here in both models of obesity investigated, we surmise that the observed decrease in NR2B expression is due primarily to lack of leptin signaling as proposed previously by other authors[61], with potential consequences also on 2-AG levels and CB1 activation. Lack of leptin in ob/ob mice might cause both the reduction of NR2B expression and the enhancement of OxA (and hence 2-AG/CB1) signaling in the DG, whereas leptin desensitization in the hypothalamus of HFD mice will only cause the latter of these two events, and CB1 antagonism in vivo (but not in vitro) would also only block the latter one[42]. As a consequence, and as observed by us, NR2B expression is expected to be reduced much more in the DG of ob/ob mice (due to concomitant reduction of leptin signaling and increase of OxA levels), and much less in HFD mice (where there is only the increase in OxA levels), and the higher DG levels of 2-AG in HFD mice (and hence the much lower counteraction by AM251 of impaired LTP in vivo in these mice), would be explained with the well-known contribution of NMDAR activation to 2-AG biosynthesis/release[55–57].

Altogether, our data suggest that prolonged consumption of diets with high-fat content, and subsequent development of obesity, might have a significant impact on cognitive functions by promoting neurochemical changes affecting glutamate release and signaling. Indeed, we observed a strong impairment in neurogenesis-dependent LTP in obese mice when this was recorded in vivo, in line with the behavioral alterations observed during the PS test and according to others[18,19], but not ex vivo in slices. This apparent discrepancy may be explained by the loss or dilution of molecular factors in the ACSF medium during ex vivo experiments[43]. OxA, a water-soluble neuropeptide that regulates not only learning and memory processes but also AHN by stimulating cell differentiation, is very likely to be one such factor[39], by virtue of its effects on hippocampal synaptic plasticity in the DG and CA1 areas[40,70,71]. Accordingly, OxA signaling at Ox-1R, in vivo, and in vitro[42], often requires eCB signaling at CB1 receptors, which in turn is known to promote and regulate AHN[29,30], whereas overactivation of CB1 has been reported to induce deficits in memory processes[35]. Starting from previous evidence showing that OxA release is increased in several output areas of the LH[42], here we demonstrated that OxA is over-released to the DG where, via downstream elevation of 2-AG levels and activation of CB1 receptors, it inhibits MPP plasticity, thus leading to impaired episodic memory and PS. Accordingly, recovery of both defective LTP and PS impairment in obese mice was obtained following blockade of Ox-1R and/or CB1 receptors.

If the ACSF-LTP (i.e. without GABA blockers in the bath) observed in hippocampal brain slices is exclusively attributable to immature GCs[15,72], the same effect in in-vivo recordings could be contributed by mature GCs, according to Drew et al.[73]. Thus, in obese mice, eCB signaling overactivity following increased OxA release into the DG, would, on the one hand, increase the number of mature neurons and, on the other hand, impede their participation in MPP-DG LTP, which is crucial for the establishment of PS[15–17]. This paradoxical phenomenon would explain why we observe impaired MPP-DG LTP in obese mice even in the presence of enhanced neuronal differentiation. Wang et al. demonstrated that 2-AG mediates LTP at lateral perforant path-DG (LPP) excitatory synapses[34,74], while it does not affect the LTP of the MPP-mature granule cell synapses in in-vitro slices from lean mice[74]. For this reason, we speculate that the inhibition of the MPP-DG LTP observed in vivo in obese mice is attributable primarily to excessive OxA/eCB signaling in the DG of obese mice, combined with a neurogenesis-dependent effect absent in lean mice. In this sense, our present findings extend our knowledge regarding memory-related plasticity, by providing evidence for a form of OxA/eCB/CB1-mediated inhibition of LTP of cortical inputs to the hippocampus, regulating "what" information is processed in the episodic memory and "where"[75]. Indeed, inhibition of LTP via CB1 is well documented in Shaffer collateral synapses[76], and it is possible that OxA/eCB/CB1-dependent inhibition of MPP-DG LTP, with subsequent impairment of PS, represents a maladaptive response to OxA/eCB/CB1-induced altered AHN in the DG of obese mice. Such aberrant response is clearly different from, for example, physical exercise-induced neurogenesis, which has been previously found to be associated with both physiological elevations of hippocampal eCB signaling and amelioration of cognitive functions[77,78].

A central question arises concerning how leptin, OxA, and 2-AG affect the regulation of AHN in obese mice. In this regard, it is well known that 2-AG promotes adult neurogenesis in normal standard fat-fed mice, by acting in paracrine and/or autocrine manner at the CB1 receptors to increase the number of BrdU-positive cells[32,79]. Furthermore, CB1 receptors and 2-AG have been shown to stimulate neurogenesis from neural stem cells in the adult mouse hippocampus, with consequences on neural plasticity and learning and affective functions[80,81], whereas CB1-deficient mice display reduced adult hippocampal neuronal

precursor proliferation[82]. Chronic administration of leptin to adult mice increases cell proliferation without significant effects on the differentiation and survival of newly proliferated cells in the DG where expression of the long-form of the leptin receptor, LepRb, has been detected in hippocampal progenitor cells[83]. Finally, Ox-A has been found to promote neurodifferentiation of the newly proliferated cells by enhancing the number of DCX/BrdU- and NeuN/BrdU-positive nuclei in the DG[39]. In line with this effect, chronic treatment with the dual orexin receptor antagonist almorexant neither did increase hippocampal cell proliferation nor neurogenesis[84]. We have observed here a decrease of proliferation and an increase in DCX+ and NeuroD+ cells, together with revealing increased neuronal morphology, which, based on this previous literature, could be the result of: (1) impaired leptin signaling and/or CB1 desensitization due to the Ox-A-mediated enhancement of 2-AG levels in the DG of obese mice, in the case of reduced proliferation, or (2) increased Ox-A signaling, either per se or via increased CB1 activation by 2-AG, in the case of enhanced formation of the neuronal precursors.

In summary, our study suggests a mechanism through which the OxA/2-AG/CB1 pathway influences AHN and synaptic plasticity of the DG, thereby affecting PS processing and episodic memory. The fact that this pathway, which requires to be tightly regulated in time and space, may participate in both beneficial adaptive and negative maladaptive responses, as it is the case here for the optimal control of adult neurogenesis and memory, represents yet another example of the well-recognized pro-homeostatic function of orexinergic and, particularly, eCB signaling[85].

Possible future optogenetic and/or chemogenetic approaches could be useful to determine the functional impact of the modulation of OxA neurons on episodic memory in obese mice. Overall, uncovering how adult-born GCs function and contribute to behavior will lead to therapeutic pharmacological targets to treat cognitive impairment. The role in hippocampal neuro-transmission modulation, and hence memory, of NR2B-containing NMDARs expressed in GCs, which have been previously implicated in obesity,[28,86] should also be further explored in future studies.

## Methods

**Animals.** The study has been performed according to the ARRIVE Guidelines to improve the reporting of bioscience research using laboratory animals. Experiments were performed following the European Union animal welfare guidelines [European Communities Council Directive of September 22, 2010 (2010/63/EU)] and the Italian Decree n.26/2014, authorization n. 152/2020-PR and 589/2018.

The experiments were performed on 12–13-week-old mice. All the mice were housed in controlled temperature (20–23 °C) and humidity conditions (55 ± 5%) and fed ad libitum. Male mice with spontaneous nonsense mutation of the ob gene for leptin (ob/ob, JAX mouse strain) B6. Cg-Lepob/J, and WT ob gene expressing homozygous siblings were obtained by breeding ob gene heterozygotes and by genotyping with PCR. For the HFD experimental group, WT-ob homozygous were fed with a D12492 Research diet (60% kcal from fat) for 8 weeks starting from 1 month of life. For an experimental subset proopiomelanocortin (POMC)-eGFP mice (JAX mouse strain, C57BL/6J-Tg(Pomc-EGFP)1Low/J) were used.

Since basal orexin levels exhibit a diurnal fluctuation (concentrations increase during the dark period or active phase (i.e. ZT13- 24) and decrease during the light period or rest phase), the animals were maintained under a 12-h light:12-h dark cycle, light on at 8:00 PM, i.e. ZT0, for the entire duration of experiments before euthanizing at ZT20-22.

**Drugs.** For Elisa, Lipid Extraction and 2-AG measurement the lean, HFD and ob/ob mice were injected i.p. with several treatments as following: OxA (Tocris, 40 μg/kg, 2 h), Leptin (Sigma-Aldrich, 5 mg/kg, 2 h), SB-334867 at doses ineffective on sleep duration and locomotion according to Morello et al.[59] (SB; Tocris; 30 mg/kg for lean and 60 mg/kg for HFD and ob/ob mice, 3 h alone or 1 h before OxA injection). Evaluation of OxA efficiency on the neuronal differentiation in DG was performed as previously described by OxA icv injection (280 pmol)[41] in lean mice, whereas ob/ob mice were injected with SB (60 mg/kg) or AM251 (3 mg/kg)[87] i.p. For the LTP experiments on slice the OxA (100 and 200 nM), SB-334867 (10 μM), AM251 (4 μM), O7460 (10 μM), ACEA (125 nM) (~20 min of preincubation) were used. For the LTP in vivo experiments, SB-334867 (0.1 nmol/0.5 μl) or AM251 (0.5 nmol/0.5 μl) were microinjected before starting the recording session.

**Immunohistochemistry.** The animals were deeply anesthetized and transcardially perfused with physiological saline (room temperature) to clear the brain tissue of blood to prevent high background histological staining. Following saline, the animals were properly perfused with fixative 4% (wt/vol) paraformaldehyde/0.1 M phosphate buffer (PB), pH 7.4 prepared shortly before the procedure (<24 h) to minimize polymerization that can affect the quality of tissue. The brains were cut with a Leica CM3050S cryostat into 10-μm-thick serial sections in the coronal plane, collected in alternate series, and processed for immunofluorescence. Anatomic comparable sections of the hippocampus were processed for multiple immunofluorescences after incubation for 1 h at room temperature in PB containing 0.3% Triton and 5% donkey serum (blocking buffer). Sections were then incubated overnight at 4 °C with the mix of primary antibodies diluted in donkey serum. The following primary antibodies were used: goat anti-OX-A (1:100; Santa Cruz, cat. no. sc-8070); guinea pig anti-VGlut1 (1:200; SYSY, cat. no. 135 304); goat anti-NeuroD (1:100; Santa Cruz, cat. no. sc-1084); rabbit anti-NMDAR2B (1:200; Abcam, cat. no. ab65783); rabbit anti-CB1 receptor antibody (anti-C-terminus 461–472, Abcam, ab23703, 1:300), goat anti-CB1 receptor antibody (anti-C-terminus 461–472, Abcam; ab40860; 1:300); rabbit anti-DCX (Abcam, ab207175; 1:300) or mouse anti-DCX (1:100; Santa Cruz, cat. no. sc-271390); mouse anti-PSD95 (1:100; SYSY, cat. no. 124011). After incubation with primary antibodies, the sections were washed with PB and immunofluorescence revealed by specific Alexa secondary donkey anti-IgGs (Invitrogen, ThermoFisher Scientific, France): alexa-350 donkey anti-goat (A21081, 1:150); alexa-350 donkey anti-rabbit (A10039, 1:100 or 1:150); alexa-488 donkey anti-rabbit (A21206, 1:150); alexa-488 donkey anti-mouse (A21202, 1:50); alexa-488 goat anti-guinea pig (A11073, 1:100); alexa-594 donkey anti-mouse (A21203, 1:50); alexa-594 donkey anti-goat (A11058, 1:50). Sections were counterstained with DAPI (Sigma-Aldrich) to detect nuclei, mounted with Prolong Gold (Invitrogen), and coverslipped with Aquatex mounting medium (Merck, Darmstadt, Germany). Controls of specificity of immunolabeling in multiple fluorescence experiments were performed by omission of primary and secondary antibodies or by preabsorption of primary antibodies with the respective blocking peptides. The immunostained sections were observed with a confocal microscopy Nikon Eclipse Ti2 (Nikon, Florence, Italy) equipped with an x-y-z motorized stage, a digital camera DS-Qi2 (Nikon, Florence, Italy), and the acquisition and Image analysis software NIS-Elements C (Nikon, Florence, Italy). Digital images were acquired using the ×20–×40 objectives. We collected serial Z-stacks of images throughout the area of interest ($n ≤ 10$ planes with an increment varying 0.5–1 μm). Images were deconvolved using the imaging deconvolution software by application of ten iterations. Serial Z planes images were collapsed into a single maximum projection image. Micrographs were saved in TIFF format and adjusted for light and contrast before being assembled on plates using Adobe Photoshop 6.01 (Adobe Systems, San Jose, CA).

**Image analysis.** For the assessment of AHN, an unbiased physical dissector-based cell counting was performed according to Zhao and van Praag[88]. Cells were counted throughout the anterior−posterior extent of the GC layer in a series of 20 μm coronal sections Ki67 or NeuroD or DCX immunolabeled and spaced 160 μm apart (1:8 as the frequency of section sampling for the cell count) starting from Bregma −1.40 mm up to Bregma −3.00 mm, to cover the complete rostrocaudal extension of the dentate gyrus. Z-stack plans of each selected section were reconstructed in 3D rendering by using the software image processing LAS-X Measurement (Leica©) for counting Ki67-, NeuroD-, DCX-ir cell number from a respective immunolabeled section. The measure of synaptic PSD95/DCX-ir or CB1/DCX-ir or OxA/DCX-ir or OxA/CB1/DCX-ir distribution was performed by applying the same criteria of image processing as above by using the respective multiple immunolabeled sections wherein, after identification of individual DCX-ir neurons and thresholding, only PSD95-ir or CB1-ir or OxA-ir or OxA/CB1-ir signal merging as pixel area were counted selectively inside the DG.

**Morphometric analysis of dendritic DXC-ir processes.** 20-μm coronal DCX-ir sections DAPI labeled were scanned on a confocal microscope Nikon Eclipse Ti2 (Nikon, Florence, Italy) equipped with an x-y-z motorized stage, a digital camera DS-Qi2 (Nikon, Florence, Italy), and the acquisition and image analysis software NIS-Elements C (Nikon, Florence, Italy). Each section was used to identify (1) the inner molecular layer (IML), which was defined as the first 50 μm from and parallel to the outer border of DAPI-positive cells, and (2) the medial molecular layer (MML), which was defined as the last 100 μm from, and parallel to, the outer border of the IML. DCX-positive dendrites within the IML and MML were imaged and level of branching was evaluated from a maximum-intensity projection of the z-series stack as described by Rosenzweig and Wojtowicz[89] and expressed as a percentage of DCX-ir area/μm$^2$ of GCL, IML, or MML including also "orphan" dendrites otherwise excluded by conventional single tracing method of dendrites. The same procedure was adopted for the quantification OxA immunoreactivity in the molecular layers.

**Electron microscopy**. Double pre-embedding immunogold labeling for correlative light and electron microscopy (CLEM) was performed in the DG (40-μm-thick) of proopiomelanocortin (POMC)-eGFP mice fixed with 3% paraformaldehyde/1% glutaraldehyde (vol/vol) in PB. An appropriate epifluorescence microscope was used to select the green fluorescent dendritic processes of POMC-eGFP newborn neurons in the IMM-ML layers in DG by being easily recognizable under appropriate excitation wavelength. The sections containing the previously selected region of ML were incubated free-floating overnight at 4 °C with the appropriate mixture of primary antibodies (rabbit anti-CB1R antibody, anti-C terminus 461–472, Abcam; goat anti- OX1R, Santa Cruz; goat anti-OxA, Santa Cruz; guinea pig anti-DAGLα, all diluted 1:50 in donkey serum blocking solution with 0.02% saponin. Subsequently, the sections were incubated in a mixture of specific gold-conjugated secondary antibodies (Aurion) carrying 10 nm metal particles to reveal OX1R or DAGLα primary antibodies or 20 nm metal particles to reveal CB1, OxA, or Ox1R. The mixture of secondary antibodies was diluted 1:30 in a donkey serum blocking solution containing 0.02% saponin. Each immunolabeled section was treated with 0.5% OsO4 in PB for 30 min at 4 °C, gently dehydrated in an ascending series of ethanol and propylene oxide, and then embedded in TAAB 812 resin (TAAB). During dehydration, the sections were treated with 1% uranyl acetate in 70% ethanol (vol/vol) for 15 min at 4 °C. Ultrathin sections (50 nm thickness) were cut by vibratome (Leica), collected on Formvar-coated, single- or multiple-slot (50-mesh) grids, and stained with 0.65% lead citrate. Electron micrographs were taken with the TEM microscope (FEI Tecnai G2 Spirit TWIN). The TEM observation was limited to series sectioned up to 0.6–0.8 μm depth from the external surface of pre-embedded immunolabeled tissue. Additional sections were processed in parallel as controls of reaction by omitting both or one of the primary antibodies from the mixture. No labeling was detected in the control material.

**Immunocytochemistry**. h-iPSC-derived NSCs obtained from healthy donors were purchased from Axol Bioscience (Cambridge, UK). Cells were seeded at a density of 100.000 cells/cm$^2$ in SureBond-coated culture vessel and cultured in a Neural maintenance medium (Axol Bioscience, Cambridge, UK), according to the manufacturer's protocol. The cultures were grown at 37 °C in a 5% CO$_2$/95% air atmosphere. For the spontaneous or stimuli mediated differentiation of iPSC-derived NPCs cells were treated with different compounds: (i) OX-A at 50, 100 or 200 nM for 30 min respectively; (ii) SB334867 at 10 μM for 45 min per se or 15 min before OX-A at 200 nM; (iii) 07460 at 10 μM for 45 min per se or 15 min before OX-A, (iv) SB334867 and 07460 mixed at 10 μM for both, 15 min before OX-A at 200 nM. The culture medium was changed every second day. After 72 h with the treatments, the cells were fixed with 4% (wt/vol) paraformaldehyde/0.1 M phosphate buffer (PB) pH 7.4 and rinsed with PB, then permeabilized with PB 0.3% Triton (10 min, 20 °C), finally incubated with Doublecortin (DCX) primary antibody at 4 °C. After 18 h, the cells were washed and revealed by a specific secondary antibody, Alexa Fluor 488 donkey anti-mouse IgG (H + L) (Invitrogen) incubated for 2 h in the dark. The nuclei were visualized by staining with DAPI (1 mg/ml; Sigma-Aldrich). Fluorescence-labeled cells were analyzed with a Leica DMI6000 fluorescence microscope equipped with a Leica DFC320 cooled digital CCD camera (Leica Microsystems). Images were digitally acquired at the same magnification and processed for fluorescence determination at the single-cell level with Meta-morph Imaging Software (Leica MetaMorph© AF).

**Orexin-A measurement in the hippocampus**. After treatments, mice were euthanized by cervical dislocation, the brains were removed and the hippocampus rapidly dissected from each brain on ice. The OxAA levels were measured with the mouse OxA ELISA kit (MyBioSource, San Diego, California, USA), following the manufacturer's instructions. Briefly, the hippocampus was rinsed with 1× PBS, homogenized in 1 ml of 1× PBS, and stored overnight at −20 °C. After two freeze-thaw cycles, performed to break the cell membranes, the homogenates were centrifuged for 5 min at 5000 × g, 2–8 °C. The supernatant was removed and assayed immediately. The absorbance of each well was measured at a wavelength of 450 nm using a 96-well microplate spectrophotometer (Multiskan GO; Thermo Scientific, Waltham, MA). This assay can detect mouse OxAA in the range of 10–480 pg/ml with a sensitivity of 1.0 pg.

**Lipid extraction and 2-AG measurement**. Tissue samples were pooled and analyzed using liquid chromatography–atmospheric pressure chemical ionization–mass spectrometry. The eCB 2-AG was extracted from tissues and then purified and quantified as previously described[90]. First, tissues were pooled and homogenized in 5 vol chloroform/methanol/ TrisHCl 50 mM pH 7.5 (2:1:1 by volume) containing 50 pmol of d5-2-arachidonoylglycerol (d5-2-AG) as internal deuterate standard. Homogenates were centrifuged at 13,000 × g for 16 min (4 °C), the aqueous phase plus debris were collected and four times extracted with 1 vol chloroform. The lipid-containing organic phases were dried and pre-purified by open-bed chromatography on silica columns eluted with increasing concentrations of methanol in chloroform. Fractions for 2-AG measurement were obtained by eluting the columns with 9:1 (by volume) chloroform/methanol and then analyzed by liquid chromatography atmospheric pressure chemical ionization-mass spectrometry (LC-APCI-MS), LabSolution Shimadzu. LC-APCI16 MS analyses were carried out in the selected ion monitoring mode, using m/z values of 384.35 and

379.35 (molecular ions +1 for deuterated and undeuterated 2-AG). Values are expressed as pmol per mg of wet tissue extracted.

**Western blotting analysis**. Each animal, previously anesthetized with isoflurane for 5 min, was decapitated and the hippocampus was quickly removed and washed twice in cold PBS (without Ca$^{2+}$ and Mg$^{2+}$, pH 7.4) and homogenized in lysis solution (1× TNE Buffer, 1% (v/v); TritonX-100, plus 1% protease inhibitor cocktail) at pH 7.4. Lysates were kept in an orbital shaker incubator at 220 × g, at 4 °C for 30 min, and then centrifuged for 15 min at 13,000 × g at 4 °C. The supernatants were transferred to clear tubes and quantified by the DC Protein Assay (cat. 5000111, Bio-Rad, Milan, Italy). Subsequently, the samples (60 μg of total protein) were boiled for 5 min in Laemmli SDS loading buffer and loaded on 8–10% SDS-polyacrylamide gel electrophoresis and then transferred to a PVDF membrane. Filters were incubated overnight at 4 °C with a rabbit polyclonal anti-NMDAR2B antibody (cat. Ab65783; Abcam, 1:1000). The monoclonal anti-tubulin (cat. T8203 Sigma-Aldrich; Milan, Italy) was used to check for equal protein loading. Reactive bands were detected by chemiluminescence by the use of using Clarity Western ECL (cat. 1705060 Bio-Rad, Milan, Italy). Images were analyzed on a Chemi-Doc station with Quantity-one software (Bio-Rad, Milan, Italy). Reagents: TNE buffer: (a) 50 mM TRIS HCl (cat. PHG0002; Sigma-Aldrich; Milan, Italy), (b) 100 mM sodium chloride (cat. S9888; Sigma-Aldrich; Milan, Italy), and (c) 0.1 mM ethylenediaminetetraacetic acid (EDTA) (cat EDS; Sigma-Aldrich; Milan, Italy). Protease Inhibitor Cocktail (cat. P8340; Sigma-Aldrich, Milan, Italy).

**Hippocampal slices preparation and local field potential (LTP) recordings**. Measurement of neurogenesis-dependent LTP in brain slices has been described previously[15,91]. Horizontal hippocampal slices were obtained from 2- and 3-month-old lean mice. The mice were anesthetized with isoflurane and decapitated. The brain was removed and immersed in a solution saturated with 95% O$_2$ and 5% CO$_2$ and composed of (mM): 87 NaCl, 25 NaHCO$_3$, 2.5 KCl, 0.5 CaCl$_2$, 7 MgCl$_2$, 25 glucose, 75 sucrose. The brains were cut into 400-μm-thick slices using a Leica VT1000 S Vibrating blade microtome. Slices were incubated at 37 °C for 1 h and stored at RT in solution for recording (extracellular solution) composed of (mM): 125 NaCl, 25 NaHCO$_3$, 25 glucose, 2.5 KCl, 1.25 NaH$_2$PO$_4$, 2 CaCl$_2$, 1 MgCl$_2$ and saturated with 95% O$_2$ and 5% CO$_2$. Slices were transferred in a recording chamber where the bath temperature was monitored and maintained at 30 ± 1 °C. Leica DM6000 FS microscope equipped with a WAT-902H Ultimate camera was used to visualize the hippocampal slice. Field EPSPs (fEPSP) were acquired by using a MultiClamp 700B and Digidata 1440A (Axon Instruments, Molecular Devices, Sunnyvale, CA, USA) amplifier and the pClamp 10.4 software (Axon Instruments). Extracellular electrodes (1–2 MOhm) were pulled using a Sutter P-1000 puller and filled with the extracellular solution. A monopolar glass electrode connected with an isolated pulse stimulator AM 2100 (A-M Systems, Carlsborg, WA) positioned into the MPP was used for the extracellular stimulation. Paired-pulse (PP) protocol was applied to test the correct positioning of the stimulation electrode in the MPP. The stimulation intensity was set at 50−70% of the maximal response based on the input/output relationship; the duration was 100 μs. The fEPSP slope was analyzed fitting the 20−80% of the fEPSP amplitude using Clampfit 11.1. software (Axon Instruments). The baseline for each experiment was calculated by averaging the slope values obtained after the stimulation of the MPP fibers at 0.03 Hz for 20 min. The magnitude of fEPSPs was measured in percentage to the baseline, with a stimulation frequency of 0.03 Hz for 60 min. LTP was evoked using four trains at 100 Hz, 500 ms each, repeated every 20 s. The final magnitude of LTP was determined as the average of responses between 40- and 60-min post-induction.

**In vivo recording of hippocampal long-term potentiation (LTP)**. Mice are first anesthetized with urethane (1.5 g/kg, ip) and fixed in a stereotaxic table (Stoelting Co, USA) for inserting the stimulating and the recording electrodes in the medial perforant path (MPP) and dentate gyrus (DG), respectively, following stereotaxic coordinates. The stimulating and recording electrodes are slowly lowered in the selected positions until a field excitatory postsynaptic potential (fEPSP) appears, induced by test pulses (0.2 ms in duration) delivered at the frequency of 0.033 Hz (Digitimer Ltd. model DS3). After obtaining input−output curves (I/O), generated by measuring fEPSP slope in response to increasing stimulus intensity, stable baseline responses were recorded for 15 min. Following baseline recording, a high-frequency stimulation (TBS) consisting of six trains, six bursts, six pulses at 400 Hz, interburst interval: 200 ms, intertrain interval: 20 s, is applied to MPP for the induction of LTP. After TBS, the recording of the baseline fEPSPs is continued for 60 min. The induction of LTP is considered to be successful if the amplitude and the slope of fEPSPs increase more than 20% for at least 30 min after the TBS[49,50]. To obtain fEPSP slope values, the region of interest was set from peak-to-peak, and the software (WinLTP 2.30, Bristol, UK), calculated the average slope within 10−90% of this region. Paired-pulse ratios were calculated by dividing the slope of fEPSP$_2$ by the slope of fEPSP$_1$. In LTP experiments, all data points were normalized to the average baseline slope. The recording and analysis were performed using Win LTP 2.30.

**Pattern separation during the novel object recognition test (NOR) and open-field test**. To assess learning and long-term memory, a novel object recognition (NOR) task was used. Two identical objects were placed into the arena during a 10-min sample phase. Subsequently, one of the objects was exchanged by a new object, and memory was assessed by comparing the time spent exploring the novel object as compared with the time spent exploring the familiar object in a 10-min test phase. Two weeks before the NOR experiments, the animals experienced handling by the experimenter and habituation to the arena for 5 consecutive days, respectively. The handling procedure consists of a method where simply scoops the mouse onto the hand. Using cup handling, an animal was transferred into a new cage and after a few minutes back again to the home cage. To improve the explorative behavior in obese mice, a non-aversive handling and prior familiarization with the test arena after being performed for 2 weeks. This type of handling reduced anxiety as described previously[92]. Animals that were still jumping in the NOR test were excluded from the analysis. For habituation, mice were placed into the empty arena (40 × 30 × 30 cm width × length × height, PVC) for 60 min. For NOR experiments custom-built plastic pieces (Polyoxymethylen, POM) were used with different shapes (bell: 5 cm diameter, 6 cm height, diamond: $7 \times 7 \times 7$ cm, cube $5 \times 5 \times 5$ cm) of the same color (black) or objects with different colors and shapes (lego rectangular bricks: $6 \times 4 \times 3$, cup: 6 cm diameter, 6 cm height). The objects were cleaned thoroughly with 70% ethanol followed by distilled water between trials to remove olfactory cues. During the sample phase on the first day of the NOR test, the mice were allowed to explore the two identical black objects (two bells) for 10 min. For the short-delay test phase (1.5 h) one of the sample objects was replaced by a new one with the same color but a different shape (bell by diamond) or by a new object with different color and shape (bell by lego rectangular bricks) and exploration was measured for 10 min. For the long-delay test phase (24 h) the new object was again exchanged by another new object with the same color but a different shape (diamond by cube) or by a new object with different color and shape (lego rectangular bricks by the cup).

The location of the novel object at 24 h was always different from that at 1.5 h, either first left then right, or vice versa. Consequently, the location of the familiar object also switched between the two test phases. Objects with the same color but different shapes were considered to be similar to acquisition objects.

To analyze strictly active exploration the time was measured manually using a digital stopwatch (Silva, Sweden). Active exploration was defined as direct sniffing or whisking towards the objects or direct nose contact. Climbing over the objects was not counted as exploration. The relative exploration was quantified by normalizing the difference between the exploration time of the novel (Tn) and familiar object (Tf) by the total time of exploration (Ttot) to calculate the NOR discrimination index: NOR index = (Tn – Tf)/Ttot. With identical acquisition objects, the NOR index was always less than 0.2 indicating that there was no side preference in the mice used for the study[14]. All sampling and testing trials were videotaped and analyzed using an automated behavioral tracking system (Smart v3.0, Panlab Harvard Apparatus). During the acquisition phase, two identical objects were located in the box test, was evaluated the motor activity and anxiety-like behaviors. Total distance traveled meter, frequency, and total duration of central zone visits were recorded for 10 min and analyzed.

**Water maze task**. Mice were trained in the reference memory version of the Morris water maze task to locate a hidden escape platform in a circular pool (150 cm diameter, 35.5 cm high)[48,49]. Water was made opaque with non-toxic tempera paint and kept at a temperature of (20 ± 1 °C). A white platform (37 cm long × 13 cm wide × 14 cm high) was submerged 1 cm below the water surface. Each mouse was given six trials per day for 6 consecutive days with an intertrial interval (ITI) of 30 min. Mice were released from one of four possible starting points and allowed to search up to 60 s for the platform. During the first day, the platform was visible and dislocated every two trials in a different quadrant (beacon training). From days 2 to 4, the platform was submerged in the North quadrant; during each day, the starting position remained constant (spatial training). At the end of each trial and irrespective of trial performance, mice were guided to the platform and allowed to remain there for at least 15 s. A probe trial lasting 60 s without the platform was performed before the first relearning trial on day 4. Immediately after the probe trial, the platform was displaced in the quadrant South, and animals were trained for 2 days (goal reversal training). The experimental protocol is reported in Fig. 2a. Swim paths were recorded using a video camera (PANASONIC WV-BP330) connected to a video-tracking system (ANY-MAZE 7.08, Stoelting, USA). Latency to reach the platform, path lengths, time spent in quadrants, number of goal crossings, and latency to the first entry in the goal were analyzed. Path efficiency, an index of the efficiency of the path taken by the animal to get from the first position to the first position within the goal, was also analyzed. To assess goal-related plasticity, we counted the number of trials needed by an animal to regain a path length that was smaller or at least equal to the average path length reached by that animal on day 4 at the end of the first acquisition.

**Light/dark box test**. The light−dark box (600 mm × 300 mm × 300 mm; length × width × height) apparatus consisted of two equally sized compartments: the dark compartment (black perspex) was covered, whereas the light compartment (white perspex) was open and brightly lit from above (~150 lux). Access between compartments was allowed through a partition door (70 mm × 70 mm). At the beginning of the session, mice were placed in the light compartment and were free to explore for 10 min. The time spent in each compartment, latency of first entry into the dark compartment, and the number of entries into each compartment were recorded[91]. Tests were videotaped and analyzed using an automated behavioral tracking system (Smart v3.0, Panlab Harvard Apparatus).

**Statistic**. All results were plotted in graphs and analyzed statistically using GraphPad Prism 8 (GraphPad Software, USA). In all tests, $p < 0.05$ was considered significant. Data are presented as a box plot and single data points are reported. The box plots elements are: center line, median (Q2); square symbol, mean; box limits, 25th (Q1)−75th (Q3) percentiles; whisker length is determined by the outermost data points (see supplementary information for numerical parameters). The D'Agostino−Pearson, Shapiro−Wilk, and Kolmogorov−Smirnov test were first applied to confirm the normal distribution of the data. One-way ANOVA/ Bonferroni, Kruskal–Wallis/Dunn's tests, or two-way ANOVA with Tukey post hoc were used to analyze data appropriately. In the Water Maze Task, a one-way ANOVA with diet as the main factor was performed. Effects on path length and latency during the training were assessed using repeated measurements ANOVA with the day as a second main factor.

**Reporting summary**. Further information on research design is available in the Nature Research Reporting Summary linked to this article.

## Data availability
Source data are provided with this paper.

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

## Acknowledgements

The financial support from EU, grant "PON IDF SHARID" n. ARS01_01270 - CUP B66G18000640005 and Joint International Research Unit (JIRU) for Chemical and Biomolecular Research on the Microbiome and its impact on Metabolic Health and Nutrition (MicroMeNu) is acknowledged. This study is also supported by the Sentinelle Nord Program of Université Laval (to N.F., L.C. and V.D.M.). The authors also want to thank Dr. Roberta Verde (ICB-CNR Pozzuoli, Italy) for the technical support to LC-MS mass spectrometry analysis and Dr. Elena Polishchuk (Telethon Institute of Genetics and Medicine, Pozzuoli, Italy) for technical support at ultrasectioning and TEM acquisition.

## Author contributions

N.F. designed and performed the in vivo and in vitro electrophysiological and NOR experiments, and prepared the figures. S.B. designed and performed all in vivo electrophysiological experiments. L.T., A.C.F-R. and R.I. designed and performed all biochemical, immunohistochemical, and immunocytochemical experiments and prepared the figures. F.A.I. designed and performed the western blot study. M.I. designed and performed the NOR experiments. M.D.R. and E.D.L. designed and performed the Morris Water Maze task; F.P. carried out the biochemical study by LC-MS mass spectrometry. P.D.G., R.C. and S.M. supervised pharmacological and behavioral experiments. N.F., V.D.M., and L.C. conceived the study and wrote the final version of the manuscript. L.C. supervised the work. All the authors discussed the data, edited and approved the final version of the manuscript.

## Competing interests

The authors declare no competing interests.
