## [Peer Review File · Nature Communications]

Orexin-A and endocannabinoids are involved in obesity-associated alteration of hippocampal neurogenesis, plasticity, and episodic memory in miceReviewers' Comments:

Reviewer #1:

Remarks to the Author:

In this study, Forte and colleagues report that a particular model of obesity (ob/ob) is associated with an impairment of a specific type of episodic memory known as pattern separation, by reducing hippocampal neurogenesis-mediated LTP via the orexin-A/endocannabinoid pathway. The overactivity of this pathway leads to inhibition of neurogenesis-mediated LTP in the dentate gyrus. Memory impairment was reverted by inhibiting orexin-A/endocannabinoid signaling in ob/ob mice. This is a study that describes an extremely specific mechanism in a particular genetic mouse model of obesity. The manuscript has several major limitations that difficult the interpretation of the results:

Major comments

- The most important limitation of the study is the small sample size for the behavioral studies. A sample size of n=4 for lean mice and of n=6 for ob/ob mice is too reduced in order to obtain conclusive results.
- The novelty of the study is reduced considering that previous studies using the obese leptin receptor knockout (db/db) mice also showed altered adult hippocampal neurogenesis following prolonged consumption of high-fat diet.
- The obesity model used in the study is the specific ob/ob model. The use of this specific genetic model of obesity affects the conclusions of the study because it is difficult to generalize the results obtained in this genetic model to the general obese population.

Other comments

- In figure 1d the discrimination index (DI) of lean mice is of 0.17 in the 24h long-term memory test. This is an extremely low DI compared to the one obtained in the 1.5h short-term memory test (DI=0.35). Are there significant differences between both DIs in the lean group depending on the time point of the test? Is the DI of lean mice at 24h different from acquisition (0.098 vs 0.17)? And in the ob/ob mice? The DI at 24h in the acquisition phase in figure 6b is of 0.11 and is very similar to the one obtained by lean mice at 24h in figure 1d (0.17).
- Line 148. Separate DG from We.
- In figure 3a is not clear in the X-axes if ob/ob mice have been injected with lep or SB. It is mentioned in the figure legend, but not in the text.
- In line 151, figure 3c shows NeuroD positive cells and should be cited.
- Figure 3f is not cited in the text.
- In figure 3, the concentrations of SB can be depicted as well.
- In lines 166-167, the text is referred to figure 4a and the levels of OxA but they are not represented in the figure. Instead, the levels of 2-AG are represented. It seems that figures 3a and 4a are switched.
- In line 174 it should say Figure 4a instead of figure 4b.
- In line 197, correct "ob/ob mice.. In particular".
- In figure 6d the DI of lean mice is of 0.27, which is much higher than the DI obtained in the lean mice after 24h in figure 1d (DI=0.17). This difference could be due to the small sample size.

Reviewer #2:

Remarks to the Author:

The paper by Forte et al. studies obesity-induced changes of adult hippocampal neurogenesis (AHN) and hence pattern separation in a mouse model of leptin deficiency (ob/ob mice). Observed impairments including neurogenesis-related LTP are explained by an overactive orexin-A/endocannabinoid pathway. By using an inhibitor of the OxA1 receptor the authors show reversion of the behavioral, cellular and electrophysiological phenotype. While there are interesting aspects in the study, there are significant technical issues that diminish the impact of the paper.

Comments (1-7: major)

1) The title states that adult hippocampal neurogenesis is reduced in obese mice. Yet, the authors state in the abstract and show as data an upregulation of NeuroD positive cells in the used ob/ob mouse model. This is not congruent and the title must be changed.

2) The authors used solely NeuroD IHC as a method to quantify neurogenesis. On one hand side, it is true that NeuroD is an early marker of neuronal differentiation of newborn neurons in the hippocampus. Nevertheless, the gold standard to measure adult neurogenesis is missing: Injection of BrdU and then to measure first cell proliferation and secondly the amount of new mature neurons by double staining of BrdU and NeuN. Alternatively to BrdU/NeuN staining, DCX quantification could be done. Otherwise, it is not correct to propose changes in adult neurogenesis, even if having a hint of enhanced neuronal differentiation by NeuroD staining. Furthermore, it is very difficult to discuss the results in comparison with other papers if not the same methods were applied (Ref 27-30, 43, 45).

3) A reduction of adult hippocampal neurogenesis would be in accordance with most of papers cited in the manuscript addressing the role of adult neurogenesis and obesity by high fat diet (Ref 27-29: Park et al., 2010; Murata et al., 2017; Lindqvist et al., 2006). Indeed, in the paper of Ramos-Rodriguez et al., 2014 (Ref 30), increases in cell proliferation in db/db mice were reported (leptin R deficient), but there are also other data describing AHN to be decreased in db/db mice (see e.g. Stranahan et al., 2008, Nat Neurosci). The authors need to discuss these differences in results and obese mouse models in more details. In addition, it is not discussed why in this study an AHN upregulation is observed, when leptin is missing since the authors state in the introduction that leptin increases AHN (line 67). Nonetheless, a reference demonstrating this is missing and should be added (compare Ref 24-26).

4) The authors state in the last part of the introduction (line 89/90) that LTP and PS were restored by blocking Ox1 and CB1 receptors. Even if true for LTP, AM251 was not used in the PS paradigm. Furthermore, to be consistent, the downregulation of AHN should be shown by AM251 in ob/ob mice.

5) Figure 4b: Double IHC of NeuroD/Ox-1R is not convincing. The authors need to show single channels in addition, as well as higher magnification pictures. Furthermore, it is necessary to show at which state cells start to express Ox1-R, as it is known that CB1 is already expressed at the level of neural stem cells. An anti-OxR1 double staining with BrdU or Sox2 should resolve this question. Figure 4c: Triple IHC of VGlut1/NeuroD and CB1 is not demonstrated as stated in line 182/183 that glutamatergic synapses expressing CB1 receptors contact NeuroD positive neurons. High resolution microscopy is needed at this point. Furthermore, single channels should be visible as well as magnified area indicated.

6) Regarding behavioral data: The group sizes are very low (n=4 or n=5), in particular for the control (lean). At least a repetition with another 5-6 mice must be added. Further, an additional behavioral assay would strengthen the data (please consult other work on pattern separation). Moreover, the general behavior of the mice is not shown (locomotion, anxiety, open field without objects, etc.) in order to exclude any confounding. It is known in particular that obese rodents have many behavioral alterations in comparison to lean controls.

7) The electrophysiological part is performed appropriately. Yes, at the end, it is a correlation between changes / rescues of electrophysiological features and behavior, as there are no electrophysiological interventions (DREADDs, optogenetics) altering PS. This has to be considered.

Reviewer #3:

Remarks to the Author:

Nature Communication

OBESITY IMPAIRS PATTERN SEPARATION BY REDUCING ADULT HIPPOCAMPAL NEUROGENESIS VIA THE OREXIN-A/ENDOCANNABINOID PATHWAY

By Forte et al.

Summary of the work:

The background of the paper is, that Orexin-A, a hypothalamic neuropeptide, is emerging as a regulator of neurogenesis as well as hippocampal plasticity. The authors provide in their manuscript evidence that Orexin-A neurons in the hypothalamus of leptin signaling deficient obese mice (ob/ob mice) undergo a rewiring from glutamatergic to GABAergic afferents and subsequent 2-AG/CB1-mediated disinhibition, which might trigger, in turn, excessive Orexin-A signaling in areas in which hypothalamus output terminates. Here, the authors convincingly demonstrate that, although adult hippocampal neurogenesis is enhanced in the DG of ob/ob mice as a consequence of enhanced Orexin-A release and 2-AG/CB1 activity in this area, episodic memory is compromised in these mice. Administration of Orexin-A to WT lean mice induced enhancement of neurogenesis in the dentate gyrus, whereas selective antagonism of Orexin-A 1 receptor (Ox-1R) reduced the elevated neurogenesis of ob/ob mice. Adult hippocampal neurogenesis mediated long-term potentiation (LTP) at synapses between the medial perforant-path and the dentate gyrus was inhibited, in association to alteration of pattern separation notwithstanding the increase of adult hippocampal neurogenesis in obese mice. LTP and PS were restored in ob/ob mice by blocking Ox-1R or CB1 receptors. These data indicate a causal link between memory dysfunction and Orexin-A /eCB deregulation of neurogenesis and synaptic plasticity in obesity.

Problems to address, comments:

Fig. 1e: The authors observe a higher number of NeuroD precursor neurons in the DG of obese mice with respect to lean mice (Fig. 1e). This would provide evidence that there might be more neurogenic differentiation, but not necessarily that more newly generated neurons are really surviving and integrating in the circuitry. Here the authors need to show data, that show new neurons, which are functionally integrated into the hippocampal circuitry.

(Fig. 2g, h): The authors report a difference in LTP reduction between in vivo and vitro and speculate that this suggests that alterations in the MPP-DG pathway in ob/ob mice, and the subsequent reduction of LTP, is possibly mediated by molecules in the brain which instead lost their effect during the in vitro LTP recording because of dilution in the ACSF. Here the authors should provide evidence for their claim, either with Ab stainings in slices, or by adding the missing compounds/molecules as a rescue of function experiment.

Fig. 4b/4c: The authors state that the NeuroD positive neurons in the DG express Ox-1R (Fig. 4b) and that glutamatergic synapses expressing CB1 receptors contact NeuroD positive neurons (Fig. 4c). Here a quantification is missing. In addition, this assumption seems to be based on a maximal intensity projection, this can be highly misleading if one wants to look for colocalization. Here the analysis need to be improved.

The LTP recordings in 5c are highly unstable. Why is this? I am not convinced that the quality of recording and recording conditions can be used to draw precise conclusions.

Minor:

The Graphical abstract draws some conclusions that are in my view not really addressed in the manuscript, like the role of PLC and DAGL downstream of Ox-1R. Also in the description they mention glutamate release, which comes from other papers but I think that the graphic abstract should focus on the present, new results.

Response point-to-point to the Reviewers' comments:

Reviewer #1 (Remarks to the Author):

In this study, Forte and colleagues report that a particular model of obesity (ob/ob) is associated with an impairment of a specific type of episodic memory known as pattern separation, by reducing hippocampal neurogenesis-mediated LTP via the orexin-A/endocannabinoid pathway. The overactivity of this pathway leads to inhibition of neurogenesis-mediated LTP in the dentate gyrus. Memory impairment was reverted by inhibiting orexin-A/endocannabinoid signaling in ob/ob mice. This is a study that describes an extremely specific mechanism in a particular genetic mouse model of obesity. The manuscript has several major limitations that difficult the interpretation of the results:

Major comments

- The most important limitation of the study is the small sample size for the behavioral studies. A sample size of n=4 for lean mice and of n=6 for ob/ob mice is too reduced in order to obtain conclusive results.

As requested by the reviewer we increased the number of mice to n=10 mice for each experimental group. The new results are depicted in Fig. 1 and Fig. 7 of the revised paper and throughout the text of the "Results" section.

- The novelty of the study is reduced considering that previous studies using the obese leptin receptor knockout (db/db) mice also showed altered adult hippocampal neurogenesis following prolonged consumption of high-fat diet.

We regret for not having highlighted enough the strength and novelty of our study showing that it is the overactivity of orexinergic/endocannabinoid signaling that leads to inhibition of AHN-mediated long term potentiation (LTP) in the DG with subsequent impairment of pattern separation and its functional impact on episodic memory in obese *ob/ob* leptin knockout mice. Although the quoted reference n.30 (Ramos-Rodriguez *et al. PloS one* 2014; 9,e89229, that is reported as n.33 in the revised paper) and the one suggested by the reviewer #2 (Stranahan AM *et al. Nat Neurosci.* 2008; 11:309–317) are referred to *db/db* leptin receptor knockout mice, that are similar to *ob/ob* mice in as much as they lack leptin signalling, the *db/db* model of obesity exhibited a reduction, rather than an increase, of OX-A levels in the lateral hypothalamus (see Gonzales and Prehn, *Neuroscience* 2018;369:183-191 and Mondal *et al., Regul Pept.* 2002;104:21-25). In this sense, these previous reports are completely different from our report in the genetically *ob/ob* and in high-fat diet obese mice.

- The obesity model used in the study is the specific ob/ob model. The use of this specific genetic model of obesity affects the conclusions of the study because it is difficult to generalize the results obtained in this genetic model to the general obese population.

Taking advantage from this important point raised by the reviewer we were able to enrich the translational value of the work by extending the study to the High Fat Diet-fed group of mice (HFD; #D12492 Research Diet ©) with 60% of kilocalories from the saturated fats. In the revised paper we present data from behavioural, anatomical and electrophysiological experiments carried out by comparing age-matched genetically obese *ob/ob* mice to HFD-fed group and wt control mice fed with a standard-fat diet (SFD). These results are reported throughout in the text and in the Figs.1; 2; 3E-H; 4A,F,H-J; 5A,C; 6B,E,G,H; 7; Suppl.Fig.1 of the revised paper.

Other comments

- In figure 1d the discrimination index (DI) of lean mice is of 0.17 in the 24h long-term memory test. This is an extremely low DI compared to the one obtained in the 1.5h short-term memory test (DI=0.35). Are there significant differences between both DIs in the lean group depending on the***

time point of the test? Is the DI of lean mice at 24h different from acquisition (0.098 vs 0.17)? And in the ob/ob mice? The DI at 24h in the acquisition phase in figure 6b is of 0.11 and is very similar to the one obtained by lean mice at 24h in figure 1d (0.17).

Based on the suggestion of the reviewer, we have increased the number of lean mice to n=10. In the current study, we have observed different discrimination index (DI) values between lean and obese mice at both short- and long-term retention phase following acquisition (**Fig. 1** of the rebuttal letter). In particular, we found in lean mice a DI of 0.31 ± 0.03 (1.5 h) and 0.25 ± 0.03 (24 h) that were significantly different with respect to the acquisition phase (0.08 ± 0.04) (Kruskal-Wallis with Dunn's post hoc; Kruskal-Wallis statistic=8.516) (**Fig. 1A** of the rebuttal letter). On the contrary, ob/ob mice showed, in the long-term retention phase, a DI (-0.20 ± 0.12) significantly lower than the acquisition phase (0.1 ± 0.04). Whereas, no difference was recorded in DI between 1.5 h (0.01 ± 0.05) and acquisition phase (**Fig. 1B** of the rebuttal letter) (One Way ANOVA and Bonferroni Post Hoc test, $F = 13.58$). No differences were observed comparing the DI calculated during the acquisition phase and at 1.5h/24h in HFD mice (**Fig. 1C** of the rebuttal letter). The new DI values are reported in lines 118-135 in the revised paper. We thank the reviewer for this suggestion since the new obtained data provided a more clear portrait of the alteration in the short-term and long-term memory process in obese mice.

Figure 1. (Rebuttal Letter)

- Line 148. Separate DG from We.

The text is changed in the revised paper and line 148 doesn't report the same concept now .

- In figure 3a is not clear in the X-axes if ob/ob mice have been injected with lep or SB. It is mentioned in the figure legend, but not in the text.

We are sorry but the revised paper refers to this data in the Fig. 4a wherein the X-axes reports all the treatments. The SB was not injected in this set of experiment.

- In line 151, figure 3c shows NeuroD positive cells and should be cited.

The text is changed in the revised paper and line 151 doesn't report the same concept. Data concerning NeuroD immunohistochemistry are depicted in Suppl. Fig.2 g, h and described in lines 206-219.

- Figure 3f is not cited in the text.

We are sorry this mistake. This figure is renumbered in revised paper as Fig. 4b and data are described in line 245.

- In figure 3, the concentrations of SB can be depicted as well.

Overall, to avoid overloading of the figures we never reported the concentrations of the substances if not referred to OxA treatment for which different concentrations were used in the same experiment. Only in this case we have reported the OxA concentration in the corresponding graph (see **Fig 4b.d** as well as **Fig. 6a e Suppl.3** of the revised paper) to help the reader to better understand the OxA dose-dependent effects. In any case, if should this still is a request from the referee, in this second round of revision we have no difficulty in adapting the graphs according to the new indications.

- In lines 166-167, the text is referred to figure 4a and the levels of OxA but they are not represented in the figure. Instead, the levels of 2-AG are represented. It seems that figures 3a and 4a are switched.

We are sorry for that; in the revised version the OxA levels are reported in Fig. 4a (line 238) and the 2-AG levels are represented in Fig 5a (lines 275-277).

- In line 174 it should say Figure 4a instead of figure 4b.

This line was referred to 2-AG levels that are now represented in **Fig 5a** (line 275-277).

- In line 197, correct "ob/ob mice.. In particular".

The text is changed in the revised paper; the line 197 doesn't match with the same concept.

- In figure 6d the DI of lean mice is of 0.27, which is much higher than the DI obtained in the lean mice after 24h in figure 1d (DI=0.17). This difference could be due to the small sample size.

We agree with the reviewer. This discrepancy has been solved. The NOR DI in Fig.1d is NOR DI=0.25±0.03 during 24h long-term memory test because of the increase of the size sample (see the lines 131-134 of the revised paper). Old Figure 6d of the old text is now Fig. 7d and NOR DI=0.29±0.06 (line 364-368 in the revised text).

Reviewer #2 (Remarks to the Author):

The paper by Forte et al. studies obesity-induced changes of adult hippocampal neurogenesis (AHN) and hence pattern separation in a mouse model of leptin deficiency (ob/ob mice). Observed impairments including neurogenesis-related LTP are explained by an overactive orexin-A/endocannabinoid pathway. By using an inhibitor of the OxA1 receptor the authors show reversion of the behavioral, cellular and electrophysiological phenotype. While there are interesting aspects in the study, there are significant technical issues that diminish the impact of the paper.

Comments (1-7: major)

1) The title states that adult hippocampal neurogenesis is reduced in obese mice. Yet, the authors state in the abstract and show as data an upregulation of NeuroD positive cells in the used ob/ob mouse model. This is not congruent and the title must be changed.

There was a (rather silly) mistake in the title. We apologize with the reviewer, the title has been changed in the current revised paper.

2) The authors used solely NeuroD IHC as a method to quantify neurogenesis. On one hand side, it is true that NeuroD is an early marker of neuronal differentiation of newborn neurons in the hippocampus. Nevertheless, the gold standard to measure adult neurogenesis is missing: Injection of BrdU and then to measure first cell proliferation and secondly the amount of new mature neurons by double staining of BrdU and NeuN. Alternatively to BrdU/NeuN staining, DCX quantification could be done. Otherwise, it is not correct to propose changes in adult neurogenesis, even if having a hint of enhanced neuronal differentiation by NeuroD staining. Furthermore, it is very difficult to discuss the results in comparison with other papers if not the same methods were applied (Ref 27-30, 43, 45).

In the revised paper we have applied NeuroD (Fig.4k-l; Suppl.Fig.2a-f) and DCX immunohistochemistry (Fig.2e-g; Fig.4e-g; Fig 5b-d; Fig.6d.f) and performed appropriate quantification for Ki67 (Fig.2a), DCX (Fig.2d and Suppl.Fig.3) and NeuroD (Fig.2c and Suppl. Fig.2) in the DG of lean and obese (ob/ob and HFD) mice to address the AHN. The resulting data are reported from lines 140-146 for Ki67, 146-150 for DCX and 151-155 for NeuroD, and in particular in the section of “ Results” lines 137-174, 206-245 and 246-270.

The utility of Ki-67 as proliferative markers of adult neurogenesis is well recognized by being a nuclear protein marker expressed in all phases of the cell cycle except the resting one and usefully adopted as an endogenous marker of proliferation in the DG by mimicking the expression pattern of BrdU. Furthermore, a quantitative comparison of BrdU and Ki-67-positive cells showed 50% higher numbers of the latter when examined 24 h after the BrdU injection. This was expected since BrdU can be incorporated into DNA only during the S-phase of the mitotic process, whereas Ki-67 is expressed for its whole duration. Thus, Ki-67 is a well-recognized effective mitotic marker and has most of the benefits of BrdU and none of the costs accordingly to Kee et al., 2002 (*J. Neurosci. Methods*, for review). The changes in adult neurogenesis have now been documented by the data which support enhanced neuronal differentiation in the DG of obese mice by counting appropriately the cells expressing NeuroD (a marker of neuroblasts expressed before the doublecortin DCX) and DCX, a marker of newly generated immature neurons.

By using these markers for the study of proliferation (Ki67-ir or BrdU) and neurodifferentiation (NeuroD and DCX) in the DG is possible to confirm that a reduction in the expression of a marker of proliferation is observed in the DG of HFD mice according to Park et al., 2010 (previous ref. n.27 and n.30 in the revised paper) and Lindqvist et al., 2006 (previous ref. n.29 and n. 32 in the revised paper). Concerning data by Murata et al., 2017 (previous ref. n.28 and n. 31 in the revised paper) the discordance observed in the reduction of the number of DCX-ir cells in DG of HFD mice could be attributed to the ~ 35% lower rate of obesity obese mice recruited in the Murata’s study in comparison to our study which mice were recruited if exhibiting at least ~ 50% increase of the original weight (see Cristino et al., PNAS 2013 for details). Finally, Ramos-Rodriguez et al., 2014 (previous ref. n.30 and n. 33 in the revised paper) report increase in neuro-proliferation and neuro-differentiation in the DG of db/db mice that differ from ob/ob and HFD- mice because of the lower OxA levels in the brain.

Furthermore, according to Zhao et al., 2014 (previous ref. n.43 and n. 46 in the revised paper) and Ito et al., 2008 (previous ref. n.45 and n. 48 in the revised paper) we found that OxA exogenously administered is able to promote neurodifferentiation in DG (see Suppl. Fig.2 in the revised paper)

3) A reduction of adult hippocampal neurogenesis would be in accordance with most of papers cited in the manuscript addressing the role of adult neurogenesis and obesity by high fat diet (Ref 27-29: Park et al., 2010; Murata et al., 2017; Lindqvist et al., 2006). Indeed, in the paper of Ramos-Rodriguez et al., 2014 (Ref 30), increases in cell proliferation in db/db mice were reported (leptin R deficient), but there are also other data describing AHN to be decreased in db/db mice (see e.g. Stranahan et al., 2008, Nat Neurosci). The authors need to discuss these differences in results and obese mouse models in more details. In addition, it is not discussed why in this study an AHN upregulation is observed, when leptin is missing since the authors state in the introduction that leptin increases AHN (line 67). Nonetheless, a reference demonstrating this is missing and should be added (compare Ref 24-26).

We thank the reviewer for this criticism. In the revised paper the study of AHN has been significantly implemented revealing changes in the proliferative (i.e. number of Ki67-positive cells) and neurodifferentiative (number of DCX cells) phases. The papers quoted by the reviewer [Ramos-Rodriguez et al. (*PLoS one* 2014; 9,e89229), and Stranahan et al. (*Nat Neurosci.* 2008;11:309–317) show data about db/db leptin receptor knockout mice. These mice are similar to the ob/ob mice since they also lack leptin signalling. However, they are quite different in terms of OX-A levels in the hypothalamus, with db/db mice exhibiting a reduction of OX-A levels vs age-matched wild type littermates (see Gonzales and Prehn, *Neuroscience.* 2018;369:183-191 and Mondal et al., *Regul Pept.* 2002;104:21-25), and ob /ob mice instead

presenting, like high fat diet-obese mice, with an enhancement OX-A levels in the LH and LH output areas (Mondal et al., *Regulatory Peptides* 2002; Cristino et al., *PNAS* 2013; Backer et al., *Mol Met* 2016). Given the fact that the mechanism we report is dependent on increased OX-A and, hence, endocannabinoid signalling, this may explain why AHN upregulation is observed in ob/ob conversely to db/db mice (as reported in the Introduction). Indeed, ob/ob and db/db mice present with several other differences, and in fact are used as models of obesity and type 2 diabetes, respectively.

Concerning the request of the reviewer to discuss why in our study an AHN upregulation is observed in concomitance with lacking of leptin signalling (line 67 that is line 64 in the revised paper) we can argue that this effect, in both ob/ob and HFD mice, could be due to the “inverted-U-shaped dose-related function” of leptin as reported by Garza et al 2008 *J.Biol Chem*. In these terms, low or high levels of leptin could not positively affect the DG cell proliferation, learning and memory function according to Weyner et al., 2004 (previous ref. n.46 and n. 49 in the revised paper) and Oomura et al., 2006 (previous ref. n.26 and n. 29 in the revised paper). In this case we can ascribe the AHN upregulation in DG of ob/ob and HFD as effect of the increased OxA levels in the brain as showed in Fig 2, Suppl. Fig 2). Fig.6b,c (functional effect on PS) Fig. 7 (behavioural effect on PS) Suppl. Fig.2 and Suppl. Fig.3.

4) The authors state in the last part of the introduction (line 89/90) that LTP and PS were restored by blocking Ox1 and CB1 receptors. Even if true for LTP, AM251 was not used in the PS paradigm. Furthermore, to be consistent, the downregulation of AHN should be shown by AM251 in ob/ob mice.

We apologize for this mistake, which has been corrected in the revised paper by rewording the part of the “Introduction” from lines 82-90. To meet the reviewer’s request we now demonstrated in Supplementary Figure 2f,h of the revised paper that CB1 antagonism by AM251 prevents the enhancement of AHN as also reported in the text from lines 209-219.

5) Figure 4b: Double IHC of NeuroD/Ox-1R is not convincing. The authors need to show single channels in addition, as well as higher magnification pictures. Furthermore, it is necessary to show at which state cells start to express Ox1-R, as it is known that CB1 is already expressed at the level of neural stem cells. An anti-OxR1 double staining with BrdU or Sox2 should resolve this question. Figure 4c: Triple IHC of VGlut1/NeuroD and CB1 is not demonstrated as stated in line 182/183 that glutamatergic synapses expressing CB1 receptors contact NeuroD positive neurons. High resolution microscopy is needed at this point. Furthermore, single channels should be visible as well as magnified area indicated.

According to the request of this reviewer to show at which state the DG cells start to express Ox1-R we performed Ox1-R double immunostaining with Ki67 or NeuroD or DCX. The results showing the presence of Ox1-R starting from the proliferative (Ki67-positive cells) phase of AHN up to the neuro-differentiative one (NeuroD- or DCX-positive cells) are depicted in the Fig. 2 of this rebuttal letter. The DCX/Ox1-R immunostaining data are reported in the submitted revised paper (Fig. 5E,G; lines 293-295 and lines 298-300) by performing the high-resolution correlative light and electron microscopy.

Figure 2. (Rebuttal Letter)

Our revised paper reports that over-differentiation of immature GCs from NeuroD (a marker of neuroblasts) to DCX (a marker of newly immature neurons) alters the synaptic plasticity of MPP-DG synapses via 2-AG-mediated CB1 receptor overactivation. Altogether, the present results are focussed on the morphological study of MPP/immature dentate granule cells GCs (i.e. DCX cells) synapses; most of the data in the paper are referred to the DCX immunostaining accordingly. The triple IHC of VGlut1/NeuroD/CB1 depicted in the previous Fig.4c is now reported in Fig.4k,l of the revised paper, and single channels are shown in Fig. 3 of this rebuttal letter.

Figure 3. (Rebuttal Letter)

According to the need raised by the referee to apply high-resolution microscopy to characterize the CB1 expressing glutamatergic inputs to DG neurons involved in the MPP-DG plasticity we have performed a high-resolution correlative light and electron microscopy. The cellular target is DCX-positive instead of NeuroD-positive by being the DCX cells exclusively involved in the functioning of the MPP-DG pathway that regulates the PS.

At this purpose, we exploited the ML-DG area of POMC-eGFP mice which represent a widely recognized model of eGFP ectopic labeling of DCX-like newborn neurons of the DG (Overstreet et al., 2004; Journal of Neuroscience).

6) Regarding behavioral data: The group sizes are very low (n=4 or n=5), in particular for the control (lean). At least a repetition with another 5-6 mice must be added. Further, an additional behavioral assay would strengthen the data (please consult other work on pattern separation). Moreover, the general behavior of the mice is not shown (locomotion, anxiety, open field without objects, etc.) in order to exclude any confounding. It is known in particular that obese rodents have many behavioral alterations in comparison to lean controls.

As suggested by the reviewer we have now increased the size of the sample by adding another 5-6 mice per group to reach an average size of n=10 mice/experiment. Furthermore, by taking advantage of a similar issue raised by reviewer #1 we were able to enrich also the group of mice by including the group of mice made obese by feeding high-fat diet (HFD) as reported in Figs.1; 2; 3E-H; 4A,E,F,H-J; 5A,C; 6B,E,G,H; 7; Suppl.Fig.1 of the revised paper. Of note, accordingly to this reviewer's suggestion, we have performed the light/dark box and open field on lean, HFD and ob/ob mice showing that after 2 weeks of habituation the ob/ob mice have improved all the parameters analyzed in the open field test like the traveled distance, time in the center and visit the central area which support an overall reduction anxiety and increment in the locomotion activity as reported in the Supplementary figure 1.

7) The electrophysiological part is performed appropriately. Yes, at the end, it is a correlation between changes / rescues of electrophysiological features and behavior, as there are no electrophysiological interventions (DREADDs, optogenetics) altering PS. This has to be considered.

Reviewer #3 (Remarks to the Author):

Nature Communication

OBESITY IMPAIRS PATTERN SEPARATION BY REDUCING ADULT HIPPOCAMPAL NEUROGENESIS VIA THE OREXIN-A/ENDOCANNABINOID PATHWAY

By Forte et al.

Summary of the work:

The background of the paper is, that Orexin-A, a hypothalamic neuropeptide, is emerging as a regulator of neurogenesis as well as hippocampal plasticity. The authors provide in their manuscript evidence that Orexin-A neurons in the hypothalamus of leptin signaling deficient obese mice (ob/ob mice) undergo a rewiring from glutamatergic to GABAergic afferents and subsequent 2-AG/CB1-mediated disinhibition, which might trigger, in turn, excessive Orexin-A signaling in areas in which hypothalamus output terminates. Here, the authors convincingly demonstrate that, although adult hippocampal neurogenesis is enhanced in the DG of ob/ob mice as a consequence of enhanced Orexin-A release and 2-AG/CB1 activity in this area, episodic memory is compromised in these mice. Administration of Orexin-A to WT lean mice induced enhancement of neurogenesis in the dentate gyrus, whereas selective antagonism of Orexin-A 1 receptor (Ox-1R) reduced the elevated neurogenesis of ob/ob mice. Adult hippocampal neurogenesis mediated long-term potentiation (LTP) at synapses between the medial perforant-path and the dentate gyrus was inhibited, in association to alteration of pattern separation notwithstanding the increase of adult hippocampal neurogenesis in obese mice. LTP and PS were restored in ob/ob mice by blocking Ox-1R or CB1 receptors. These data indicate a causal link between memory dysfunction and Orexin-A /eCB deregulation of neurogenesis and synaptic plasticity in obesity.

Problems to address, comments:

- **(Fig. 1e): The authors observe a higher number of NeuroD precursor neurons in the DG of obese mice with respect to lean mice (Fig. 1e). This would provide evidence that there might be more neurogenic differentiation, but not necessarily that more newly generated neurons are really surviving and integrating in the circuitry. Here the authors need to show data, that show new neurons, which are functionally integrated into the hippocampal circuitry.**

Quantification of Ki67, NeuroD, and DCX immunoreactive cells has been performed and integration of adult-born neurons (that are revealed by DCX expression) has been studied by coupling DCX with the marker of synaptogenesis PSD95, thus providing evidence that more newly generated neurons are present in DG of obese mice (both ob/ob and HFD), and more are integrated into the MPP-DG circuitry,

- **(Fig. 2g, h): The authors report a difference in LTP reduction between in vivo and vitro and speculate that this suggests that alterations in the MPP-DG pathway in ob/ob mice, and the subsequent reduction of LTP, is possibly mediated by molecules in the brain which instead lost their effect during the in vitro LTP recording because of dilution in the ACSF. Here the authors should provide evidence for their claim, either with Ab stainings in slices, or by adding the missing compounds/molecules as a rescue of function experiment.**

According to the referee's suggestion, we supported our speculation by performing *ex vivo* LTP recordings by adding the missing compounds/molecules in a rescue of function experiment. In particular were added different OxA concentrations (see Fig. 4b-d, and line 191-205 of the revised text), and the endocannabinoid like 2-AG compound ACEA (see Fig. 6a and line 308-317 of the revised text) to demonstrate that OxA and 2-AG are able to inhibit LTP, thus recapitulating the results obtained *in vivo*.

- **(Fig. 4b/4c): The authors state that the NeuroD positive neurons in the DG express Ox-1R (Fig. 4b) and that glutamatergic synapses expressing CB1 receptors contact NeuroD positive neurons (Fig. 4c). Here a quantification is missing. In addition, this assumption seems to be based on a maximal intensity**

projection, this can be highly misleading if one wants to look for colocalization. Here the analysis need to be improved.

Our revised paper reports that over-differentiation of immature GCs from NeuroD (a marker of neuroblasts) to DCX (a marker of newly immature neurons) alters the synaptic plasticity of MPP-DG synapses via 2-AG-mediated CB1 receptor over-activation. Altogether, the present results are focussed on the morphological study of MPP/immature dentate granule cells GCs (i.e. DCX cells) synapses; most of data in the paper are now referred to DCX immunostaining accordingly. The triple IHC of VGlut1/NeuroD/CB1 depicted in the previous Fig.4c is now reported in Fig.4k,l of the revised paper. Furthermore, the morphological study of the MPP-DG circuit and the molecular characterization of the DCX-ir newborn neurons receiving CB1- and OxA-expressing terminals has now been completely revised by exploiting high resolution scanning laser confocal microscopy with 3D deconvolution processing and by the high resolution of correlative light and electron microscopy (see fig.5b-h in the revised text).

-The LTP recordings in 5c are highly unstable. Why is this? I am not convinced that the quality of recording and recording conditions can be used to draw precise conclusions.

This figure 5c is now the Fig. 6a in the revised paper. We performed other experiments to strengthen our results and excluding the most unstable recordings.

Minor:

The Graphical abstract draws some conclusions that are in my view not really addressed in the manuscript, like the role of PLC and DAGL downstream of Ox-1R. Also in the description they mention glutamate release, which comes from other papers but I think that the graphic abstract should focus on the present, new results.

We apologize with the reviewer; we have now reorganized the graphical abstract according to the revised text.

Reviewers' Comments:

Reviewer #1:

Remarks to the Author:

The manuscript has been greatly improved in this revised version and all the major concerns have been addressed. The new experiments have provided a reasonable number of mice in each experimental group and the results are now convincing. The inclusion of the high fat diet-fed group has now added data that can be more easily generalized to the general obese population. I do not have any additional concern on this revised version of the manuscript.

Reviewer #2:

Remarks to the Author:

In the revision, the authors used Ki67 and DCX immunostaining to analyze AHN (together with NeuroD as in the previous submission). They observed a decrease of proliferation and an increase in DCX+ and NeuroD+ cells, together with revealing increased neuronal morphology. How can the authors explain the discrepancy between proliferation and differentiation? - In particular, since it is known that endocannabinoids usually upregulate proliferation, and have an AHN impact rather on stem cell proliferation than on stem cell differentiation (see in e.g., Aguado et al., 2005, 2007), whereby, however, increases of neuronal morphological attributes are indeed eCB-related. The authors must discuss this point.

If Ki67 gives more or less the same information as BrdU, does it mean that in obese mice more apoptosis of proliferating cells occurs or do we see an earlier differentiation? Normally, less proliferation means less net neurogenesis. Therefore, I doubt it is possible to speak from enhanced neurogenesis (as written in the paper, e.g., in Legend to Figure 2, line 514), if the effect in proliferation is not there or no net increase of BrdU+/NeuN+ positive cells is observed. Nevertheless, it is appropriate to talk about enhanced neuronal differentiation or alteration of AHN, but the authors should be consistent throughout in the paper.

Regarding point 6 of the previous review: An answer / action is missing on "Further, additional behavioral assays would strengthen the data (please consult other work on pattern Separation).

The answer to point 7 is missing. At least discuss the Limitation of the study is requested.

Reviewer #3:

Remarks to the Author:

Review Forte et al.

The authors addressed carefully either in the manuscript or in the response to the reviewers the mayor concerns.

In particular, the low number of animal for the behavioural studies was increased to 10 – according to our own statistical power assessment this is now sufficient if one considers the normal error margin.

This additional, but mandatory effort is well acknowledged.

The authors also convincingly argued that their mouse models and treatments are novel and significantly different from e.g db/db mice. This is well written in the rebuttal letter, but I still believe it would help the reader to make an explicit statement about this issue in the discussion.

In order not to depend to heavily on just one genetic model, the authors also added a HFD group of mice in figures 1-7 as an additional group. This is well taken and also increases the translational value of the study.

Also the title was corrected and now fits the scope of the study.

Neurogenesis has been explored now in much more detail with adequate markers (Fig. 2,4,5,6). I now agree with the conclusion of the authors. The authors also include new and much more precise data

about the presence of Ox1-R from the proliferative phase of AHN.

Finally the quality of the LTP in figure 5c and 6a is not satisfactory and exclude now unstable and unreliable recordings.

The authors have invested a huge amount of work and this improved the paper significantly and I have no further suggestions for experiments or controls, this is now an important contribution which should be mad accessible to a broad readership.

Martin Korte

Point-by-point (in blue) response to the Reviewers for the manuscript entitled: “Orexin-A and endocannabinoids are involved in obesity-associated alteration of hippocampal neurogenesis, plasticity and episodic memory” NCOMMS-20-16730A-Z

Revision changes are highlighted in blue in the revised manuscript.

We thank the Reviewers for the high appreciation of the novelty, quality, and relevance of our work. We have done our best to address with additional behavioral experiments the issues raised by Reviewer #2. We sincerely think that the revised version is improved and more straightforward and hope that it is ready for publication in Nature Communications.

REVIEWER COMMENTS

Reviewer #1 (Remarks to the Author):

The manuscript has been greatly improved in this revised version and all the major concerns have been addressed. The new experiments have provided a reasonable number of mice in each experimental group and the results are now convincing. The inclusion of the high fat diet-fed group has now added data that can be more easily generalized to the general obese population. I do not have any additional concern on this revised version of the manuscript.

The authors would like to thank the Reviewer for his/her positive comments.

Reviewer #2 (Remarks to the Author):

In the revision, the authors used Ki67 and DCX immunostaining to analyze AHN (together with NeuroD as in the previous submission). They observed a decrease of proliferation and an increase in DCX+ and NeuroD+ cells, together with revealing increased neuronal morphology. How can the authors explain the discrepancy between proliferation and differentiation? - In particular, since it is known that endocannabinoids usually upregulate proliferation, and have an AHN impact rather on stem cell proliferation than on stem cell differentiation (see in e.g., Aguado et al., 2005, 2007), whereby, however, increases of neuronal morphological attributes are indeed eCB-related. The authors must discuss this point.

We are grateful to the Reviewer for raising this point. A central question in the field of adult neurogenesis is how the multitude of signals in the neurogenic niche regulates neuronal precursor cells proliferation and/or differentiation. 2-AG, OxA, and leptin are key molecules involved in this context and their levels are altered in DG of obese, ob/ob and HFD, mice. This point has been extensively discussed in the revised text (lines 507-524) and important new references have been added (lines 1223-1235).

If Ki67 gives more or less the same information as BrdU, does it mean that in obese mice more apoptosis of proliferating cells occurs or do we see an earlier differentiation? Normally, less proliferation means less net neurogenesis. Therefore, I doubt it is possible to speak from enhanced neurogenesis (as written in the paper, e.g., in Legend to Figure 2, line 514), if the effect in proliferation is not there or no net increase of BrdU+/NeuN+ positive cells is observed. Nevertheless, it is appropriate to talk about enhanced neuronal differentiation or alteration of AHN, but the authors should be consistent throughout in the paper.

*We agree with the Reviewer and apologize to him/her for this inconsistency that has been solved throughout the revised paper and accordingly in the legend of Figure 2, which is now **Figure 3** in the revised paper (line 592).*

Regarding point 6 of the previous review: An answer / action is missing on "Further, additional behavioral assays would strengthen the data (please consult other work on pattern Separation).

*We thank the Reviewer for this comment, and we apologize for not deepening this issue in the previous revision. To avoid the use of behavioral tasks relying on the explicit negative (i.e. electric foot shock) or positive (i.e. juice, food) rewards to motivate the animals, whose processing might be non-specifically biased by obesity and endocannabinoids, respectively, we used a modified version of the object recognition task (NOR) that has been previously shown to be correlated with adult DG neurogenesis. In particular, animals are required to discriminate a new object from a familiar one, which has a similar shape and a similar color. Under these conditions, we report that obese mice are impaired both at short- and long-retention intervals. The Reviewer is correct that we did not report in the previous version of the manuscript the behavior of obese mice in a control task not requiring pattern separation. Therefore, in this revised version of the manuscript, we have included an additional group of obese and lean mice subjected to the classical version of the NOR, where the new object is completely different from the familiar one in shape and color, and found no deficits in obese mice at either delay. To further confirm these findings in an episodic memory task, as suggested by the Reviewer, we have tested an additional group of obese and lean mice in a modified version of the water maze test that has been shown to be sensitive to increased (after exercise training) and to genetic constitutive ablation (Cyclin D2 knockout mice) of adult DG neurogenesis (Garthe et al., 2014, Genes Brain Behav 13, 357–364). Obese mice did not show sensorimotor impairment while navigating toward a visible platform, by using a beacon strategy. Furthermore, they can learn to localize the quadrant in the pool where the hidden platform is located, during the spatial version of the task. However, they show a less precise knowledge of the exact location of the platform (annulus crossing) as compared to lean mice, and when the hidden platform is moved to a different quadrant, they take more time to reach it with direct paths. This type of deficit nicely recapitulates the behavioral deficits of Cyclin D2 knockout mice in the same task, thus further suggesting a deficit in pattern separation necessary to process and update detailed episodic representations. The new results are reported in the revised text (lines: 123-174), in the new panels e-g of **Fig. 1** (caption at lines 577-578) and in the new added **Fig.2** (caption at lines 577-578).*

The answer to point 7 is missing. At least discuss the Limitation of the study is requested.

We thank the Reviewer for this suggestion that is now discussed in the revised text (lines 531-534).

Reviewer #3 (Remarks to the Author):

Review Forte et al.

The authors addressed carefully either in the manuscript or in the response to the reviewers the mayor concerns. In particular, the low number of animal for the behavioural studies was increased to 10 – according to our own statistical power assessment this is now sufficient if one considers the normal error margin. This additional, but mandatory effort is well acknowledged. The authors also convincingly argued that their mouse models and treatments are novel and significantly

different from e.g db/db mice. This is well written in the rebuttal letter, but I still believe it would help the reader to make an explicit statement about this issue in the discussion. In order not to depend too heavily on just one genetic model, the authors also added a HFD group of mice in figures 1-7 as an additional group. This is well taken and also increases the translational value of the study.

Also the title was corrected and now fits the scope of the study. Neurogenesis has been explored now in much more detail with adequate markers (Fig. 2,4,5,6). I now agree with the conclusion of the authors. The authors also include new and much more precise data about the presence of Ox1-R from the proliferative phase of AHN.

Finally the quality of the LTP in figure 5c and 6a is not satisfactory and exclude now unstable and unreliable recordings.

The authors have invested a huge amount of work and this improved the paper significantly and I have no further suggestions for experiments or controls, this is now an important contribution which should be made accessible to a broad readership.

The authors would like to thank the Reviewer for his/her positive comments. The novelty and significance of the mouse models of obesity used in the study have been further emphasized in the revised text by highlighting the differences with the db/db mouse model (lines 432-441).

Reviewers' Comments:

Reviewer #2:

Remarks to the Author:

There are no further comments. The authors responded very well to the concerns posted, with even additional experiments.